# ROPES: ROBOTIC POSE ESTIMATION VIA SCORE-BASED CAUSAL REPRESENTATION LEARNING

## ABSTRACT

Causal representation learning (CRL) has emerged as a powerful *unsupervised* framework that can (i) disentangle the latent generative factors underlying high-dimensional data, and (ii) learn the cause-and-effect interactions among the disentangled variables. There have been extensive recent advances in the identifiability aspects of CRL, accompanied by some practical progress. However, a substantial gap remains between theory and real-world practice. This paper takes a step toward closing that gap by bringing CRL into robotics, a domain that has motivated CRL. Specifically, this paper addresses the well-defined robot pose estimation – the recovery of position and orientation from raw images – by introducing **RO**botic **P**ose **E**stimation via **S**core-Based CRL (**ROPES**). Being an *unsupervised* framework, ROPES embodies the essence of interventional CRL by identifying those generative factors that are actuated: images are generated by intrinsic and extrinsic latent factors (e.g., joint angles, arm/limb geometry, lighting, background, and camera configuration) and the objective is to disentangle and recover the controllable latent variables, i.e., those that can be directly manipulated (intervened upon) through actuation. Interventional CRL theory establishes that variables that undergo variations induced by interventions can be identified. In robotics, such interventions arise naturally by commanding actuators of various joints and recording images under varied controls. Empirical evaluations in semi-synthetic manipulator experiments demonstrate that ROPES successfully disentangles latent generative factors with high fidelity with respect to the ground truth. Crucially, this is achieved by leveraging only distributional changes, without using any labeled data. The paper also includes a comparison with a baseline based on a recently proposed semi-supervised framework. This paper concludes by positioning robot pose estimation as a near-practical testbed for CRL.

## 1 INTRODUCTION

Causal Representation Learning (CRL) has emerged as the confluence of three primary research directions: disentangling generative factors embedded in high-dimensional data, causal inference, and representation learning (Schölkopf et al., 2021). CRL's objective is to leverage the raw, high-dimensional observations (e.g., images, signals, or text) and perform two tasks: (i) disentangle the latent generative factors of the data, and (ii) learn the causal interactions (influence) among these variables. Causal interactions are modeled as causal mechanisms which are stable conditional probability laws that relate causes to their effects. A true causal representation would capture these stable causal mechanisms as the underlying generative processes. Learning representations that are robust, interpretable, and reliable is deemed critical for downstream reasoning and decision-making.

There have been substantial recent advances in understanding the identifiability limits of CRL, providing a clearer understanding of when and how the underlying causal factors of complex systems can be reliably recovered from data (Varıcı et al., 2025; Ahuja et al., 2023; Yao et al., 2025; Ng et al., 2025). These theoretical insights have been complemented by practical algorithms that can scale to high-dimensional settings, broadening CRL's impact across a broad spectrum of applications (Tejada-Lapuerta et al., 2025; Sun et al., 2025; Yao et al., 2024; Lee, 2024). Among these, robotics has emerged as a particularly relevant domain, where the ability to disentangle generative factors (various causal mechanisms) and capture cause-and-effect relationships in the robot's pose and its interaction with objects is central to perception, control, and decision-making. Robots operate in dynamic and

uncertain environments, where robust and interpretable representations of the world are critical for tasks such as navigation, manipulation, and interaction. As a result, robotics has motivated many of the key research questions in CRL (Schölkopf et al., 2021).

The problem of robot *pose* estimation from images naturally aligns with the CRL framework. Reliable knowledge of a robot's configuration, known as *pose*, is critical for a vast range of tasks, from robotic manipulation/control to safe human-robot interaction. The objective of pose estimation is to use sensory data (image) to recover the robot's position and orientation in 3D space.

**A generative viewpoint on pose.** Pose estimation can be naturally framed from a generative viewpoint: denote the variables that shape the pose by $Z$. These variables undergo a complex transformation (image rendering) to generate image data $X$, formalized by mapping $X = f(Z)$, where $f$ is unknown. The latent variables $Z$ are the robot's pose parameters (the Cartesian coordinates of the joints and/or joint angles determining position and orientation). Since the arm/limb lengths are invariant, tracking pose can be abstracted by tracking the variations in the joint angles. Hence, by modeling joint angles as the latent generative factors embedded in a larger generative process, we can ask whether and how these variables can be recovered without explicit labels. Furthermore, when performing any specific task, the joint angles do not vary independently, as kinematic and task constraints couple them, so that a change in one joint angle inevitably imposes changes in a subset of the rest. Such changes induce *causal* interactions among the joint angles. Hence, having latent generative factors that exhibit causal interactions renders pose estimation a problem of CRL.

The question we answer in this paper is: *Can we recover the joint angles (that form the pose) from high-dimensional observations (camera images of a robot arm)* ***without direct supervision, even for a single image?*** This perspective offers a path towards label-free pose estimation that leverages recent algorithmic advancements and formal identifiability results in CRL. To provide context, we first overview our methodology, and then discuss the relevant ML-based and model-based approaches to pose estimation, which rely on either labeled data or engineering cues about joint angles, respectively.

**Interventions via controllable variables.** In CRL literature, it is established that without *statistical diversity* or *induction bias* in the observed data, recovering the latent variables $Z$ from $X$ under an unknown transformation $X = f(Z)$ is impossible (Locatello et al., 2019). This is even true for independent component analysis (ICA), a special case with statistically independent latents (Hyvärinen & Pajunen, 1999). One effective way to achieve statistical diversity is through *interventions*, which enable identifiability of latent causal factors, even when $f$ is unknown and highly complex (Varıcı et al., 2025; von Kügelgen et al., 2023). An *intervention* applies a localized, *distribution-level* change to the latent data-generating mechanism. In robotics, interventions can be realized by grouping data collected under different actuation policies into different datasets. Each such control protocol yields a dataset whose distribution differs from others in ways that reflect the changes in the altered, or *intervened*, latent factors. A growing body of work shows that suitably designed interventional collections identify the intervened latents (up to well-understood ambiguities) (Ahuja et al., 2023; Squires et al., 2023; Varıcı et al., 2024; 2025; Buchholz et al., 2023; von Kügelgen et al., 2023; Liang et al., 2023; Yao et al., 2025; Li et al., 2024; Zhang et al., 2023; Ng et al., 2025). Thus, our goal is to recover the controllable variables (joint angles) manipulated by actuator-based interventions.

Interventions constitute a very weak form of supervision: rather than the common notion of supervision with per-sample labels, this paradigm requires only distribution-level contrasts, e.g., image sets collected under different generative regimes. While most CRL results specify when identifiability is possible, they often remain theoretical or assume conditions impractical for complex domains. Among the algorithmic frameworks that handle general transformations, score-based CRL (Varıcı et al., 2025) is a principled and practical approach: it avoids restrictive assumptions on the latent causal model and offers provable recovery guarantees, making it a natural fit for the pose estimation problem.

**Methodology.** We propose **RO**botic **P**ose **E**stimation via **S**core-Based Causal Representation Learning (**ROPES**) to learn a disentangled representation of a robot arm's state from images. We note that ROPES is domain-agnostic and it does not rely on prior knowledge of the robot's physical model, configuration, or sensing pipeline. To perform ROPES, we first collect interventional data by changing the distribution of one joint at a time, then apply a three-stage pipeline (see Figure 2). Building on the score-based CRL framework presented in Varıcı et al. (2025), we leverage the sparsity of score function differences across interventions, where a score function of a distribution is defined as the logarithm of its probability distribution. The inverse mapping for taking the observable data

Figure 1: Conceptual overview of ROPES, highlighting its three-stage pipeline. The output visualization marks the specific joints targeted for intervention, along with their respective axes of rotation.

back to the latent space is performed by a convolutional autoencoder. Recovering the causal factors in the latent space relies on the signals embedded in score differences, which are estimated using a classifier-based score estimator. Finally, a second autoencoder refines the initial latent encoding into the latent joint angles using a regularizer that implicitly constraints score functions in latent space to have sparse variation upon intervention. This method requires only distribution-level contrasts between a set of images before and after an intervention on a single joint.

**ML-based Pose Estimation (Supervised Deep Learning).** Recent advances most relevant to this work center on **supervised** deep methods that infer pose directly from images, often with substantial labeled data, and sometimes depth data or 3D computer-aided design (CAD) models. Among them, DREAM (Lee et al., 2020) frames the task as 2D keypoint detection and learn belief maps for joint locations; RoboPose (Labbé et al., 2021) uses an iterative refinement strategy to minimize prediction error on joint angles; HPE (Ban et al., 2024) has a cnn based encoder trained with ground truth labels and RoboPEPP (Goswami et al., 2025) combines a powerful pretrained encoder (I-JEPA (Assran et al., 2023)) with supervised regression. These approaches can be highly accurate with sufficient labels. However, as supervised methods, they are sensitive to shifts in distribution from the labeled domain, occlusions, and modeling assumptions (e.g., reliance on depth). Reliance on specific conditions, especially, limits their generality: models trained for one workspace or lighting regime often degrade elsewhere, and bridging that sim-to-real gap remains an active challenge (Ordoumpozanis & Papakostas, 2025; Chen et al., 2022). Our framework in this paper shows how to obtain interpretable and identifiable pose variables from images without *any* per-sample supervision.

**Model-based Pose Estimation.** More conventional solutions to pose estimation include model-based pipelines, such as using fiducial marker systems, geometric, and sensor-fusion methods. For instance, ArUco (Garrido-Jurado et al., 2014)) attaches fiducial markers to robot joints and locates them in pixel space to estimate the joint positions. These methods are precise under controlled conditions but degrade with occlusion, calibration errors, or marker loss (García-Ruiz et al., 2023).

**Contributions.** This paper bridges the theory-practice gap in interventional CRL by applying it to a well-defined robotics problem. The existing studies on CRL adopt stylized settings that do not fully reflect the reality and complexity of real-world complexities, a concern recently highlighted by Gamella et al. (2025). For generating our data, we use a widely-used experimental platform (Panda-Gym (Gallouédec et al., 2021)) that produces realistic, high-dimensional images of robotic arms under diverse actuation regimes. We show that CRL can recover joint angles, establishing that these methods can scale to visually rich and structured domains. In summary, our contributions are:

- *Formalization:* We formalize pose estimation as a CRL problem in which robot joint angles are treated as controllable latent causal variables embedded in a larger generative mapping.

- *Methodology:* We propose ROPES, an autoencoder-based architecture augmented with interventional regularizers that rely on score variations upon interventions. This relies on score-based CRL algorithms that are shown to have provable identifiability.

- *Empirical validation:* Through the experimental platform, we work with a manipulatable multi-joint robot and collect visual data. We show a strong correlation between the angles recovered by ROPES and the ground truth values.

- *No reliance on pose labels:* Our work shows disentanglement by exploiting distributional changes and therefore requires no conventional supervision from pose labels. Importantly, the algorithm is domain-agnostic and unsupervised (except for the intervention/dataset labels).

- *Comparison with state-of-the-art:* We demonstrate that ROPES, without using any pose label, achieves comparable performance to state-of-the-art RoboPEPP, which uses a JEPA-based (Assran et al., 2023) self-supervised backbone followed by supervised training to predict joint angles.

Specifically, our ablation study shows that RoboPEPP requires a substantial amount of labeled data to outperform our completely label-free method.

## 2 PROBLEM SETTING: CRL FOR ROBOTIC POSE ESTIMATION

**Pose estimation as CRL.** The pose of a robot is specified by its joint angles and joint positions. Accordingly, pose estimation aims to recover the pose from images of the robot. The images, in principle, are generated by a mapping from an array of latent factors, including the pose variables as well as other intrinsic and extrinsic factors such as arms' lengths, camera position, lighting, and background. To place the emphasis on pose estimation and disentangle it from other factors, we consider a setting in which only the joint angles vary, while all other generative factors are fixed. Formally, denote the $d$-dimensional movable joint angles of a robotic arm by $Z \triangleq [Z_1, \ldots, Z_d]$ and denote the image captured by a camera mounted at a fixed position by $X \triangleq [X_1, \ldots, X_n]$. The imaging rendering process is specified by $f$, i.e., $X = f(Z)$. Variations in the joint angles $Z$ are generally not independent. Performing a specific task imposes a structure on the robot's movements, which in turn induces dependence among the joint variables. We adopt a causal model to capture the potential interactions among the joint variables. The combination of causal interactions in the latent space and the complex mapping from the latent to the observable space renders pose estimation as a *causal representation learning* problem.

**Latent causal generative model.** To formalize the latent and observed data models, denote the probability density functions (pdfs) of $Z$ and $X$ by $p$ and $p_X$, respectively. Following causal Bayesian network formalism (Pearl, 2009), we assume that the distribution of $Z$ factorizes with respect to a directed acyclic graph (DAG) $\mathcal{G}$ on $d$ nodes, where node $i \in [d]$ of $\mathcal{G}$ represents $Z_i$. Directed edges of $\mathcal{G}$ specify the cause-effect relationships as follows: an intervention on variable $Z_i$ (e.g., changing its statistical distribution or even fixing it to a specific joint angle) influences a change in the descendant of $Z_i$ in graph $\mathcal{G}$, while the non-descendant variables remain intact. Hence, generation of $Z_i$ is governed by the conditional distribution $p_i(z_i \mid z_{\mathrm{pa}(i)})$, where $\mathrm{pa}(i)$ denotes the set of parents of node $i$ in $\mathcal{G}$. This conditional distribution is often referred to as the causal mechanism of $Z_i$. Given $\mathcal{G}$, subsequently, the distribution of $Z$ factorizes according to $p(z) = \prod_{i=1}^{d} p_i(z_i \mid z_{\mathrm{pa}(i)})$. As standard in the CRL literature, we assume that $n \geq d$ and $f$ is differentiable. CRL's objective is to use samples of $X$ and recover the latent variables $Z$ and the causal graph $\mathcal{G}$.

**Interventions.** Viability of CRL hinges on having a form of statistical diversity in the samples of $X$ (see Locatello et al. (2019); Yao et al. (2025); Komanduri et al. (2024) for discussions). An effective mechanism of inducing the needed diversity is through *interventions*. Intervention on variable $Z_i$ means changing its generating mechanism $p_i(z_i \mid z_{\mathrm{pa}(i)})$ to another conditional distribution. We perform single-joint interventions by manipulating one joint angle independently at a time while allowing variations of other joint angles to be distributionally similar to the pre-manipulation data. This new set of poses forms the interventional dataset. Such interventions are realistic in robotic systems, since we can typically manipulate a specific joint angle independently using actuators.

In this paper, we consider *stochastic hard interventions* as the most commonly studied type of intervention in the CRL literature (von Kügelgen et al., 2023; Buchholz et al., 2023; Varıcı et al., 2025; Squires et al., 2023). A hard intervention on variable $Z_i$ removes the effects of its parents and replaces the causal mechanism $p_i(z_i \mid z_{\mathrm{pa}(i)})$ with a distinct mechanism $q_i(z_i)$. Under this intervention, the joint distribution of $Z$ changes from $p$ to $q^i$, which factorizes according to $q^i(z) = q_i(z_i) \prod_{i \neq j}^{d} p_j(z_j \mid z_{\mathrm{pa}(j)})$. Except for the knowledge that some specific joint has been distributionally intervened on, **we require no pose labels**.

**Objective.** Formally, the objective is to recover joint angles $Z$ from interventional images generated by intervening on all or a subset of the joint angles without requiring any explicit pose annotation. In our approach, given an interventional dataset for joint $i$, we aim to learn a mapping $h_i : \mathbb{R}^n \to \mathbb{R}$ such that $h_i(X)$ is only a function of the angle of joint $i$, i.e. $Z_i$, by using the original random set of poses and the intervened set of random poses considered as observational and interventional distributions.

## 3 SCORE-BASED METHODOLOGY: ROPES

Our design of ROPES is grounded in the score-based algorithms (Varıcı et al., 2025); however, substantial refinements were required to operationalize the theoretical framework in this application. Next, we provide an overview of ROPES, describe the implementation steps in detail, and describe the data generation model.

### 3.1 Score-Based Interventional CRL: Key Properties

In this subsection, we review the key properties of the score functions and their variations that are instrumental for designing ROPES. Score function of the pdf $p$ is defined as $s(z) \triangleq \nabla_z \log p(z)$. We use two hard stochastic intervention mechanisms for variable $Z_i$, denoted by $q_i(z_i)$ and $\bar{q}_i(z_i)$. We require $q_i$ and $\bar{q}_i$ to be sufficiently different in distribution, formally defined via *interventional discrepancy* (Liang et al., 2023), stating that the two interventions have different statistical imprints.

**Assumption 1** (Interventional Discrepancy). $\nabla_{z_i} \log \frac{q_i(z_i)}{\bar{q}_i(z_i)}$ *is nonzero almost everywhere (i.e., it can only vanish on a set of Lebesgue measure zero).*

Next, for a particular joint of interest $Z_i$, we create a pair of interventional images using the same camera and two post-intervention $q^i$ and $\bar{q}^i$. Denote the score functions of these interventional distributions by $s_q^i, s_{\bar{q}}^i : \mathbb{R}^d \to \mathbb{R}^d$. When comparing these two interventional distributions, we observe that the score difference is a one-sparse vector in coordinate $i$, i.e., $\mathbb{E}\left[\left|s_q^i(z) - s_{\bar{q}}^i(z)\right|\right]_j \neq 0 \iff j = i$. Built on this observation, it can also be readily shown that the representation (or a coordinate-wise scaled version of it) is the unique minimizer of the following loss (and it attains 0)

$$\mathcal{L} = \left\| \mathbb{E}\left[\left|s_q^i(z) - s_{\bar{q}}^i(z)\right|\right] - e_i \right\|_2^2 , \tag{1}$$

where $e_i$ is the standard $d$-dimensional unit vector[1]. To operationalize this observation, however, we need a mechanism that can compute the score difference in the latent space, noting that we do not have direct access to the realizations of the latent variables. To this end, we leverage the following connection between score differences in the latent and observable spaces (Varıcı et al., 2025, Lemma 8),

$$s_q^i(z) - s_{\bar{q}}^i(z) = [J_f(z)]^\top \cdot \left[s_q^i(x) - s_{\bar{q}}^i(x)\right] , \quad \text{where } x = f(z) , \tag{2}$$

where $J_f(z)$ denotes the Jacobian of $f$ at point $z$. Given these relationships, we first find the score difference in the image space. Subsequently, for any choice of an encoder-decoder pair $(h, g)$, where encoder $h$ is the mapping from observation to latent space and decoder $g$ is the reverse, the loss function has two pieces. One component enforces reconstruction by the $(h, g)$ pair, and the second one promotes the sparsity structure specified by (1). The aggregate loss is formalized as follows:

$$\mathcal{L}(h, g) = \underbrace{\mathbb{E}\left[\|g \circ h(x) - x\|^2\right]}_{\text{Reconstruction Loss}} + \lambda \underbrace{\left\| \mathbb{E}\left[\left|J_g(\hat{z})^\top \cdot (s_q^i(x) - s_{\bar{q}}^i(x))\right|\right] - e_i \right\|^2}_{\text{Sparsity Loss}} . \tag{3}$$

The next theorem establishes the properties of the pair $(h, g)$ that minimizes the loss $\mathcal{L}(h, g)$.

**Theorem 1** (Theorem 22 (Varıcı et al., 2025), reworded). *Assume that the latent distribution $p$ has non-zero density over $\mathbb{R}^d$, $f$ is a diffeomorphism onto its image, and that pair $(q^i, \bar{q}^i)$ satisfies interventional discrepancy. Then, the global optimizer $(h^*, g^*)$ of $\mathcal{L}(h, g)$ recovers latent $z_i$ up to an elementwise transform, that is, $[h^*(x)]_i = \varphi(z_i)$ for some $\varphi : \mathbb{R} \to \mathbb{R}$.*

We demonstrate the efficacy of this result in a scaled-up practical problem of robot pose estimation (using data generated by robot simulators) in the rest of the paper.

### 3.2 Data Generation: Interventions via Manipulation

**Observational and Interventional Distributions.** Our inference process is unsupervised, meaning we do not require pose-labeled images. The data needed for inference are generated via interventions, which are applied by manipulating the joints individually and capturing the resulting images. Following Section 3.1, for each joint variable $Z_i$, we have one observational and two interventional distributions, $p$, $q_i$, and $\bar{q}_i$. Our learning algorithm is oblivious to any metadata or statistics associated with these distributions and learns from a random collection of poses generated before and after interventions. We denote the images drawn from $q_i$ and $\bar{q}_i$ by '0' and '1' respectively. Details of these distributions are described next, and exact parameterizations are given in Section D.

**Data Generation.** We generate our dataset using a Franka Emika Panda arm in a Panda-Gym simulator (Gallouédec et al., 2021). The arm has six primary joint angles, which are marked in Figure 1. Our data generation process consists of two stages. First, we focus on a setup with a single camera, generating interventions for the joints whose movements are confined to the camera's

---

[1] The nonzero score entry can be set to 1 by rescaling $z$: Multiplying $z$ by $c$ scales $s_q^i$ by $1/c$.

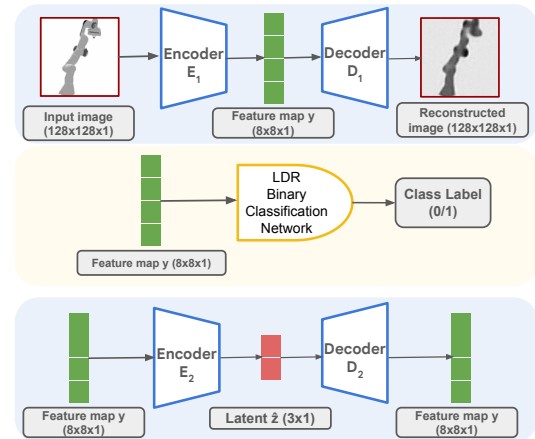

Figure 2: **Overview of ROPES pipeline.** A shared Autoencoder 1 (AE1) compresses each $128 \times 128 \times 1$ image into an $8 \times 8 \times 1$ feature map. The subsequent data processing depends on the experimental setup.

**Single-camera case** (analyzing 3 joints): feature map is directly fed into both the LDR network and Autoencoder 2 (AE2). AE2 then produces the final $3 \times 1$ disentangled pose vector.

**Two-camera case** (analyzing 6 joints): feature maps from both views are concatenated along the channel axis, forming an $8 \times 8 \times 2$ input to the LDR and AE2. AE2 then outputs the final $6 \times 1$ pose vector.

plane, i.e., the 2D projection plane, perpendicular to the camera's viewing direction. In the second stage, to accommodate the out-of-plane motions from other joints, which cannot be captured from a single viewpoint, we expand the dataset using images generated by two camera angles. The images in the dataset are converted to grayscale, resulting in a shape of $128 \times 128 \times 1$ for each image.

*Data on In-Plane Joints using a Single Camera.* We first focus on joints 2, 4, and 6, as their movement primarily results in motion within the camera's plane, and a single camera view is sufficient to cover the plane. Each data point in this dataset consists of a set of six interventional images (two for each joint) anchored by a single observational state. This observational state is generated by sampling a configuration for all six joints from a base truncated normal distribution. From this anchor, we create two distinct "hard interventions" for each of the target joints (2, 4, and 6). To perform an intervention on a specific joint, its angle is resampled from a distribution with a mean shifted far from the observational mean, while all other joint angles are held constant. This results in one observational image and six interventional images per data point.

*Data on All Joints using Two Cameras.* To extend our analysis to all six degrees of freedom, we also need to consider joints 1, 3, and 5, whose actuation causes significant out-of-plane motion. To ensure full observability, we captured every pose from two distinct camera angles ($45°$ and $135°$ yaw). We augment the dataset such that each data point now begins with an observational pose captured from both cameras (2 images). Subsequently, we perform two hard interventions on each of the six joints ($j \in \{1, \ldots, 6\}$). Each of these 12 interventional pose distributions is captured by both camera angles. Thus, a single complete data point in this extended dataset consists of 26 images: 2 observational images plus 24 interventional images (6 joints $\times$ 2 interventions/joint $\times$ 2 cameras/pose).

### 3.3 ROPES End-to-End Pipeline

The implementation of the ROPES framework consists of three stages, which we describe in this subsection. The network architectures for each of these steps are presented in Section A.

**Autoencoder-1 (AE1): Dimensionality reduction.** The first stage performs pre-processing to manage the dimensionality of the data. This is achieved by compressing the high-dimensional visual data into a lower-dimensional space. For this compression, we train a deep convolutional autoencoder, denoted by AE1, to map a grayscale image $X \in \mathbb{R}^{128 \times 128 \times 1}$ to the compressed feature map $Y \in \mathbb{R}^{8 \times 8 \times 1}$ such that $Y = E_1(X)$. The encoder ($E_1$) and decoder ($D_1$) are symmetric, featuring a multi-stage architecture with residual blocks and group normalization for stable training. AE1 is trained on the entire dataset $\mathcal{D}$ by minimizing the mean squared error (MSE) reconstruction loss:

$$\mathcal{L}_{\text{AE1}} = \mathbb{E}_X \big\| X - D_1(E_1(X)) \big\|_2^2. \tag{4}$$

The trained encoder $E_1$ serves as a fixed feature extractor for the next stage. This stage is identical for both the single-camera and two-camera cases in terms of its input, output, and latent shapes.

**Score difference estimator.** It is well-established that a binary classifier trained with cross-entropy to distinguish two distributions learns their log-density ratio (Gutmann & Hyvärinen, 2012). We leverage this principle to estimate the score difference between our two distinct interventions for each joint. For each joint $i$, we train a binary classifier, a log-density ratio (LDR) estimator, $f^i_{\text{LDR}}$, which

Table 1: MCC and MSE of ROPES across different settings. MSE is reported in radians squared.

| Model Setup | Joint 1 MCC | Joint 1 MSE | Joint 2 MCC | Joint 2 MSE | Joint 3 MCC | Joint 3 MSE | Joint 4 MCC | Joint 4 MSE | Joint 5 MCC | Joint 5 MSE | Joint 6 MCC | Joint 6 MSE |
|---|---|---|---|---|---|---|---|---|---|---|---|---|
| 1C, indep. | – | – | 0.949 | 0.053 | – | – | 0.975 | 0.029 | – | – | 0.957 | 0.049 |
| 2C, indep. | 0.874 | 0.083 | 0.979 | 0.015 | 0.634 | 0.217 | 0.950 | 0.035 | 0.679 | 0.198 | 0.884 | 0.080 |
| 2C, causal | 0.921 | 0.058 | 0.966 | 0.020 | 0.788 | 0.106 | 0.976 | 0.019 | 0.742 | 0.051 | 0.756 | 0.070 |
| 2C, indep., occl. | 0.844 | 0.101 | 0.964 | 0.025 | 0.568 | 0.245 | 0.884 | 0.082 | 0.617 | 0.225 | 0.768 | 0.145 |

1C = one camera; 2C = two cameras; indep. and causal = joints distribution; occl. = occlusion(32x32).

classifies whether a given $Y \triangleq E_1(X)$ was generated from the first ($q^i$) or second ($\bar{q}^i$) interventional distribution. After training, the gradient of the classifier's logit, $\nabla_y f^i_{\text{LDR}}(y)$, provides a direct estimate of the score difference. We note that this score difference is computed in the compressed space generated by AE1, not the original pixel space. The input to the LDR network is adapted based on the experimental setup. In the single-camera configuration, the LDR directly processes the $8 \times 8 \times 1$ compressed feature map $y$. In the two-camera case, any given "sample" yields two compressed vectors corresponding to the two camera angles. These are concatenated along the channel axis to produce a single $8 \times 8 \times 2$ input tensor for the LDR.

**Autoencoder-2 (AE2): Latent space disentanglement.** In the last stage, we train another autoencoder, denoted by AE2, with encoder $E_2$ and decoder $D_2$. AE2 is trained on the compressed data $Y$ extracted by AE1 with a loss function given in (3) to minimize the reconstruction loss and score difference sparsity loss. The input to AE2 depends on the setup: it is the $8 \times 8 \times 1$ output of AE1 for single-camera experiments, or a channel-wise concatenation of the two views, forming an $8 \times 8 \times 2$ tensor, for the two-camera experiments. We perform a hyper-parameter search over the weight $\lambda$ in (3) for the best performance. We denote the output of this encoding step by $\hat{Z} \triangleq E_2(Y)$. Theoretically, it is established that the score-based frameworks can recover the ground truth joint angle $Z_i$ uniquely, up to a *monotonic* transformation. Empirically, we observe that this transformation is well-modeled by an affine function, allowing for calibration using a small labeled dataset of ground-truth samples.

## 4 EMPIRICAL RESULTS AND ANALYSIS

In this section, we present quantitative and qualitative evaluations of ROPES across various experimental conditions, including varying latent models, camera views, occlusions, and comparison with the state-of-the-art. Exact details of the experimental setups are given in Sections A and C.

**Evaluation Metrics.** For all experiments, we evaluate disentanglement using two metrics. First, we measure the standard Mean Correlation Coefficient (MCC) (Khemakhem et al., 2020) on a separate 500-sample test set, which calculates the correlation between estimated and ground-truth latent variables. Secondly, to more directly assess the accuracy of the recovered joint angles, we train a linear regressor on 1,000 random samples to map latents to ground-truth angles and report the Mean Squared Error (MSE) on the same 500-sample test set. We repeat the entire evaluation process 15 times with different random test sets. The tables in our paper report the average MCC and MSE across these 15 runs, while the scatter plots visualize the single best run, selected by the highest MCC score. Note that ROPES does not use ground-truth pose labels during training; they are used only for the mse evaluation. Complete quantitative results are provided in Table 15. [2]. We discuss the observations of this table in the following three subsections.

### 4.1 SINGLE-CAMERA: INDEPENDENT JOINTS

We begin by evaluating the model in a single-camera setting, where joint angles are sampled *independently* according to the distributions specified in Table 6. As this setup restricts visibility, we evaluate performance only on the three in-plane joints (2, 4, and 6) discussed in Section 3.2. Table 15 (first row) shows that the MCC for ROPES is consistently at least 0.94. Remarkably, this disentanglement performance is stronger than that of the prior experiments in CRL studies in much

---

[2]The test set is sampled from In-distribution (specified in Table 7 and Table 9) for metrics reported in the paper. This is used for training as well (but train and test samples have no overlap). We also report results (Table 11 in Section I) on the test set from a different OOD distributions, which are defined in Table 8.

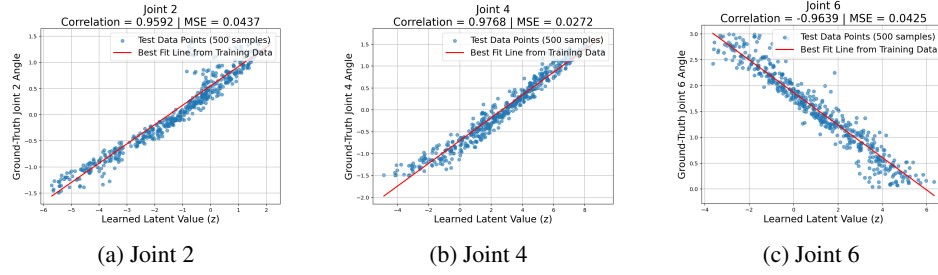

(a) Joint 2          (b) Joint 4          (c) Joint 6

Figure 3: Single Camera: Scatter-plots of ground-truth vs. estimated angles for joints 2, 4, and 6

simpler image datasets (e.g., see (Varıcı et al., 2025, Table 14)). A qualitative analysis of the model's reconstruction performance is provided in Figure 3, presenting the scatter plots of the learned latent variables against the ground-truth angles.

## 4.2 EXTENDING TO MULTIPLE VIEWS: TWO CAMERAS WITH INDEPENDENT JOINTS

We extend our analysis to a more complex two-camera setup, which provides the multi-view information necessary to disentangle all six robot joints. The joint angles are again sampled *independently* (see Table 7). As shown in Table 15 (second row) and the scatter plots in Figure 11, the model successfully disentangles all six joints. However, we observe a performance drop for joints 3 and 5 compared to the others. We attribute this performance discrepancy to less precise score estimates from the LDR network, an interpretation supported by the fact that these two joints exhibited a relatively larger classification loss during LDR training. This suggests that the images of the hard interventions for joints 3 and 5 are less distinct, making them inherently more challenging to classify. Notably, the MCC scores for joints 2, 4, and 6 remain robust when transitioning from the single-camera to the two-camera setup. The stability of these scores suggests that ROPES scales effectively, leveraging the multi-view information. Reconstructed images from the AE1 and AE2 are shown in the Figure 9.

## 4.3 TWO-CAMERA WITH CAUSAL MODEL OVER JOINTS

To assess whether ROPES can learn *causal* latents, which is the core premise of CRL, we generate a new dataset where joint angles are sampled according to a linear causal model described in Section F. The structure and edge weights were chosen randomly, with joints 1, 2, and 4 set as root nodes. We report the results in Table 15 (third row). We observe that introducing a causal data generation process generally improves performance, with MSE values decreasing across most joints compared to the independent two-camera model. Overall, this experiment demonstrates the flexibility of ROPES to accommodate causal interactions among the joints.

## 4.4 ROBOPEPP: LABEL SUPERVISION BASELINE

For our experiments, we also compare against a recent supervised baseline. Specifically, we use RoboPEPP (Goswami et al., 2025), which is among the few alternatives that can predict pose with unknown joint angles at test time (along with HPE (Ban et al., 2024) and RoboPOSE (Labbé et al.,

Table 2: MSE (in rad$^2$) for each joint comparison between ROPES and RoboPEPP trained by varying number of supervision labels and evaluated on their respective ID datasets.

| Method | Model Setup | Joint 1 | Joint 2 | Joint 3 | Joint 4 | Joint 5 | Joint 6 |
|---|---|---|---|---|---|---|---|
| RoboPEPP | 1.3K labels | 0.136 | 0.039 | 0.237 | 0.053 | 0.253 | 0.200 |
|  | 1.3K labels, occl. | 0.186 | 0.060 | 0.305 | 0.125 | 0.331 | 0.263 |
| RoboPEPP | 6.5K labels | **0.075** | 0.017 | **0.134** | **0.024** | **0.153** | 0.081 |
|  | 6.5K labels, occl. | 0.140 | 0.050 | 0.304 | 0.198 | 0.299 | 0.232 |
| RoboPEPP | 13K labels | 0.030 | 0.010 | 0.072 | 0.022 | 0.091 | 0.063 |
|  | 13K labels, occl. | 0.077 | 0.036 | 0.194 | 0.180 | 0.181 | 0.136 |
| RoboPEPP | 130K labels | 0.003 | 0.001 | 0.007 | 0.003 | 0.010 | 0.011 |
|  | 130K labels, occl. | 0.066 | 0.036 | 0.097 | 0.053 | 0.090 | 0.045 |
| ROPES | 2C, indep. | 0.083 | **0.015** | 0.217 | 0.035 | 0.198 | **0.080** |
|  | 2C, indep., occl. | **0.101** | **0.025** | **0.245** | **0.082** | **0.225** | **0.145** |
|  | 2C, causal | 0.058 | 0.020 | 0.106 | 0.019 | 0.051 | 0.070 |

1C = one camera; 2C = two cameras; indep. and causal = joints distribution; occl. = occlusion(32x32).

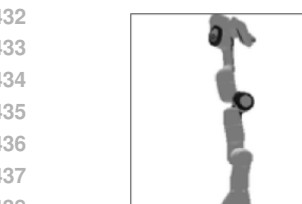 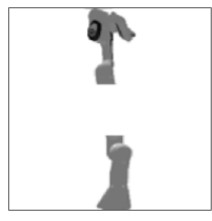 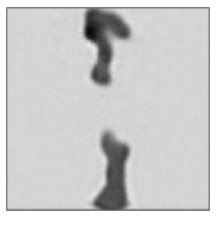 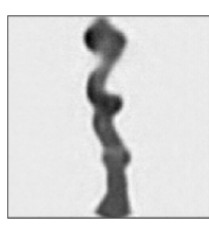

| (a) The original input image to the system. | (b) Input with $32 \times 32$ pixel white occlusion. | (c) Reconstruction from autoencoder-1. | (d) Reconstruction from autoencoder-2 |

Figure 4: A step-by-step visualization of the reconstruction process for an occluded input. The final reconstruction from autoencoder-2, generated by passing its output through the decoder of autoencoder-1, successfully inpaints the occluded region.

2021)), and reports stronger results than the alternatives. RoboPEPP involves a two-stage training process. First, a self-supervised I-JEPA backbone is pre-trained on the entire ROPES independent ID training dataset 7. In the second stage, a "JointNet" is trained to predict six joint angles. Its input is formed by concatenating the I-JEPA embeddings from two camera angles, and it is optimized using an $L_2$ loss with the ground-truth joint angle labels. We observe that the MSE of the RoboPEPP baseline decreases with increasing size of labeled data, with our model ROPES achieving a comparable performance on joints 1, 2, 4, and 6 to RoboPEPP when using labels of 5% (approx. 6.5K labels and 2 camera orientation per pose = 13K samples) of the entire dataset for training. Unlike RoboPEPP, which requires extensive training over multiple epochs and is prone to overfitting, ROPES is trained for just a single epoch, making it compute efficient.

### 4.5 ROBUSTNESS TO OCCLUSION

Finally, we evaluate ROPES' robustness to corruptions using the trained two-camera model with independent joints. At test time, we introduce artificial occlusions in the form of 32x32 white pixel squares into the input images, and report results for ROPES and RoboPEPP in Table 2. ROPES was not trained on data containing occlusions or any infilling task. Despite this, our method demonstrates superior robustness to occlusions at test time. This is evident from the MSE of joints 2 and 4, where the performance degradation for ROPES is significantly less than for RoboPEPP when occlusions are introduced. Furthermore, ROPES maintains a lower overall MSE on these joints at both 10% and 100% training data labels, highlighting its robustness to this unseen perturbation. Figure 4 visualizes the process. While the initial reconstruction from AE1 clearly shows the occluded patch, the final reconstruction from AE2 successfully inpaints the missing region by leveraging the learned disentangled representation. Table 15 (fourth row) presents the quantitative results for this condition, demonstrating strong performance despite the corruption. A detailed analysis of model performance across varying occlusion sizes is provided in Sections H and I.

We conclude that AE1 is tasked with high-quality reconstruction, while AE2's loss function sparsifies the score difference. To capture out-of-plane rotations missed by a single camera, an additional viewpoint was essential. This lowered the LDR loss cross-entropy, allowing the model to successfully differentiate all six joint movements.

## 5 CONCLUDING REMARKS

In summary, we have addressed the theory-practice gap in CRL for a robotics application, which has been a key motivating factor for CRL. Our framework extends the scope of CRL from mostly toy datasets to a semi-synthetic, close-to-real-world robotics simulator. We demonstrate that our method can recover robot joint angles, achieving a significantly high MCC and a very low MSE for many joint angles (cf. Table 15). Notably, this recovery is achieved through interventions alone, eliminating the need for explicit labels. We outperform RoboPEPP based on JEPA learning frameworks even with a substantial number of labeled images. Our framework disentangled only those joint angles whose interventions were used in the loss Eqn. (3), achieving partial disentanglement empirically at this scale. We are not aware of any larger-scale demonstration of CRL that shows such partial and incremental disentanglement when only relevant interventions/actions/changes are available.

Our work has a significant application potential in recent video world models in robotics, such as DreamGen (Jang et al., 2025). These models operate by imagining future trajectories in a high-dimensional image space, which then serves as input for a Vision-Language-Action model to predict subsequent states. Our CRL-based approach can enable direct prediction of the robot's underlying

state, i.e., its joint configuration. This dimensionality reduction from high-dimensional images to few joint angle values, can substantially accelerate the learning process for planning.

**Limitations.** We discuss some key theoretical limitations and empirical challenges that future work should take into account. First, as discussed in Section 4.1, consistent occlusion of a joint, e.g., due to a specific camera angle, can impede the recovery of the occluded joint angles, and a multi-camera setup is needed to circumvent this issue. Second, the current framework supports only single-node (i.e., single-joint) interventions. The existing CRL theory for multi-node interventions is mostly limited to linear transformations (Varıcı et al., 2024), and extending it to general transformations would help improve applicability to complex tasks such as pose estimation. Finally, we see an interesting trend in the score-difference estimator stage of the ROPES pipeline: when the interventions are starkly different, e.g., having wildly different supports, the task of the LDR estimator becomes too easy, which negatively affects disentanglement performance, since the final stage of training relies on accurate, sparse score differences. This suggests that, in practice, interventions should have at least partly overlapping support.

## ETHICS AND REPRODUCIBILITY STATEMENT

We have described our entire architecture in sufficient detail (Section A) and hyperparameters (Section C) used for our experimental results. The neural net architectures involve standard convolution filters with residual connections that can be implemented with any deep learning framework (pytorch, keras and JAX). Our work demonstrates disentangling of latent factors from high dimensional data in robot pose estimation. As such, we do not foresee any ethical concerns with methods in this work.

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

# A  ARCHITECTURE DETAILS

Table 3 details the architecture of the first autoencoder (AE1), which is identical for both the single- and two-camera experiments. The architectures for the Log-Density Ratio (LDR) network and the second autoencoder (AE2), which are adapted for each setup, are presented in Table 4 and Table 5, respectively.

Table 3: Autoencoder1 architecture ResNet-style with GroupNorm

| Component | Layer-wise Details |
|---|---|
| **Block Def.** | **ResBlockGN(f)**:
    GroupNorm → ReLU → Conv(features=f, ks=3, pad='SAME')
    → GroupNorm → ReLU → Conv(features=f, ks=3, pad='SAME')
    → Add residual input
*(Note: 'ks'=kernel size, 's'=stride, 'pad'='SAME')* |
| **Encoder** | **Input: Image** $X \in \mathbb{R}^{128 \times 128 \times 1}$
Conv(features=64, ks=3, pad='SAME')
ResBlockGN(64) x 2
Conv(features=64, ks=3, s=2, pad='SAME'), ReLU    *// Downsample 128 → 64*
Conv(features=128, ks=3, pad='SAME')
ResBlockGN(128) x 2
Conv(features=128, ks=3, s=2, pad='SAME'), ReLU    *// Downsample 64 → 32*
Conv(features=256, ks=3, pad='SAME')
ResBlockGN(256) x 2
Conv(features=256, ks=3, s=2, pad='SAME'), ReLU    *// Downsample 32 → 16*
Conv(features=512, ks=3, pad='SAME')
ResBlockGN(512) x 2
Conv(features=1, ks=3, s=2, pad='SAME'), ReLU    *// Downsample 16 → 8*
**Output: Feature map** $Y \in \mathbb{R}^{8 \times 8 \times 1}$ |
| **Decoder** | **Input: Feature map** $Y \in \mathbb{R}^{8 \times 8 \times 1}$
Conv(features=512, ks=3, pad='SAME')
ResBlockGN(512) x 2
ConvTranspose(features=512, ks=4, s=2, pad='SAME'), ReLU    *// Upsample 8 → 16*
Conv(features=256, ks=3, pad='SAME')
ResBlockGN(256) x 2
ConvTranspose(features=256, ks=4, s=2, pad='SAME'), ReLU    *// Upsample 16 → 32*
Conv(features=128, ks=3, pad='SAME')
ResBlockGN(128) x 2
ConvTranspose(features=128, ks=4, s=2, pad='SAME'), ReLU    *// Upsample 32 → 64*
Conv(features=64, ks=3, pad='SAME')
ResBlockGN(64) x 2
ConvTranspose(features=64, ks=4, s=2, pad='SAME'), ReLU    *// Upsample 64 → 128*
**Conv(features=1, ks=3, pad='SAME'), ReLU**    *// Final convolution to 1 channel*
Reshape to (batch, $128 \times 128 \times 1$)
**Output: Reconstructed Image** $\hat{X} \in \mathbb{R}^{128 \times 128 \times 1}$ |
| **Training** | Adam optimizer with learning rate = $1 \times 10^{-4}$ |

Table 4: LDR Network Architectures for Single- and Two-Camera Setups.

| Component | Layer-wise Details |
|---|---|
| **Input Processing** | The input shape depends on the camera setup:
**Single-Camera:** Input $y \in \mathbb{R}^{8 \times 8 \times 1}$ is used directly.
**Two-Camera:** Two feature maps $y$ are concatenated to form an input $\in \mathbb{R}^{8 \times 8 \times 2}$. |
| **Core Architecture** | The following layers are applied to the processed input:
Conv(features=32, ks=3), ReLU     *// Spatial dim: 8x8 → 6x6*
Conv(features=64, ks=3), ReLU     *// Spatial dim: 6x6 → 4x4*
Conv(features=128, ks=3), ReLU     *// Spatial dim: 4x4 → 2x2*
Flatten
Dense(features=128), ReLU
Dense(features=1)
**Output: Logit** $\in \mathbb{R}^1$ |
| **Training** | Adam optimizer with learning rate = $1 \times 10^{-3}$.
Minimize binary cross-entropy with logits loss on the output. |

Table 5: Autoencoder 2 (AE2) Architectures for Single- and Two-Camera Setups.

| Component | Layer-wise Details |
|---|---|
| **Block Def.** | **ResBlockGN(f)**:
GroupNorm → ReLU → Conv(features=f, ks=3, pad='SAME')
→ GroupNorm → ReLU → Conv(features=f, ks=3, pad='SAME')
→ Add residual input
*(Note: 'ks'=kernel size, 's'=stride, 'pad'='SAME')* |
| **Encoder** | **Input:** $y \in \mathbb{R}^{8 \times 8 \times C_{in}}$, where $C_{in}$ is 1 (single-cam) or 2 (two-cam).
Conv(features=64, ks=3, pad='SAME')
ResBlockGN(64) x 2
Conv(features=64, ks=3, s=2, pad='SAME'), ReLU     *// Downsample 8 → 4*
Conv(features=128, ks=3, pad='SAME')
ResBlockGN(128) x 2
Conv(features=128, ks=3, s=2, pad='SAME'), ReLU     *// Downsample 4 → 2*
Conv(features=256, ks=3, pad='SAME')
ResBlockGN(256) x 2
Conv(features=256, ks=3, s=2, pad='SAME'), ReLU     *// Downsample 2 → 1*
Flatten to (batch, 256)
Dense(features=$D_{\text{latent}}$), where $D_{\text{latent}}$ is 3 (single-cam) or 6 (two-cam).
**Output: Latent** $\hat{z} \in \mathbb{R}^{D_{\text{latent}}}$ |
| **Decoder** | **Input: Latent** $\hat{z} \in \mathbb{R}^{D_{\text{latent}}}$
Dense(features=256), ReLU
Reshape to (batch, $1 \times 1 \times 256$)
Conv(features=512, ks=3, pad='SAME')
ResBlockGN(512) x 2
ConvTranspose(features=512, ks=4, s=2, pad='SAME'), ReLU     *// Upsample 1 → 2*
Conv(features=256, ks=3, pad='SAME')
ResBlockGN(256) x 2
ConvTranspose(features=256, ks=4, s=2, pad='SAME'), ReLU     *// Upsample 2 → 4*
Conv(features=128, ks=3, pad='SAME')
ResBlockGN(128) x 2
ConvTranspose(features=128, ks=4, s=2, pad='SAME'), ReLU     *// Upsample 4 → 8*
Conv(features=$C_{in}$, ks=3, pad='SAME'), ReLU
Reshape to (batch, $8 \times 8 \times C_{in}$)
**Output: Reconstructed** $\hat{y} \in \mathbb{R}^{8 \times 8 \times C_{in}}$ |
| **Training** | Adam optimizer with learning rate = $7 \times 10^{-5}$. |

# B  DATASET FIGURES

To provide a qualitative understanding of our dataset, Figure 6 and Figure 7 visualize the hard intervention images that form the basis of our training data. Figure 6 shows intervention images for Joint 4 from the single-camera dataset. Figure 7 shows intervention images for Joint 3 from the two-camera dataset. Since Joint 3 moves out of the plane, these images highlight the necessity of our two-camera setup.

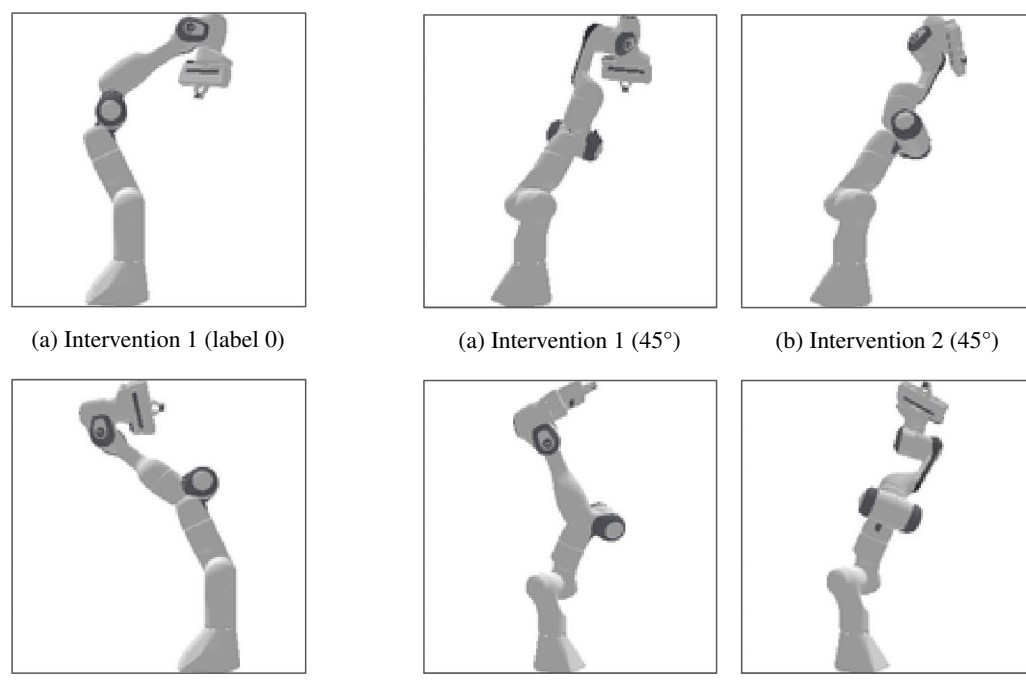

(a) Intervention 1 (label 0)          (a) Intervention 1 (45°)          (b) Intervention 2 (45°)

(b) Intervention 2 (label 1)          (c) Intervention 1 (135°)          (d) Intervention 2 (135°)

Figure 6: Two different hard interventions on Joint 4.

Figure 7: Two different hard interventions on Joint 3, each shown from two camera angles.

# C  TRAINING DETAILS

All models were trained on TPUs. We performed a hyperparameter search for the optimal learning rate, testing values in the range of $10^{-7}$ to $10^{-3}$. Our dataset consists of 10,000 observational images. As detailed in Section 3.2, we generated corresponding interventional images for two experimental setups. In the single-camera setup, each observational image yields 6 interventional images. In the two-camera setup, each observational image yields 24 interventional images. The training process involved three stages. First, Autoencoder-1 was trained on 70k images (single-camera) and 260k images (two-camera) with a batch size of 256. Second, we trained a separate binary classifier network (i.e., the log-ratio density estimator) for each joint, using 20k samples (single-camera) and 40k samples (two-camera) with a batch size of 64. Finally, for each joint, we trained a corresponding Autoencoder-2 alongside its specific LDR. Autoencoder-2 is trained on the same number of samples as Autoencoder-1 for both single camera and two camera setup.

# D  INDEPENDENT INTERVENTIONAL DISTRIBUTIONS

For the single-camera experiments, a single set of sampling distributions, detailed in Table 6, is used for both training and evaluation. As such, the concepts of in-distribution (ID) and out-of-distribution (OOD) do not apply in this case. For the more comprehensive two-camera experiments, we distinguish between ID and OOD data to evaluate model generalization. The ID dataset, which is used for training

the Independent and Occlusion models, is generated using the parameters in Table 7. To test for robustness to distributional shifts, we created a separate OOD test set with modified observational parameters, as shown in Table 8. The experiment with causal model over the joints is also evaluated on this OOD test set. We use a truncated normal distribution, denoted by $\mathcal{TN}_{[a,b]}(\mu, \sigma^2)$, which represents a normal distribution with mean $\mu$ and variance $\sigma^2$ truncated to the interval $[a, b]$.

Table 6: Sampling distributions for observational and interventional settings for single camera setup.

| Joint | Scenario | Distribution |
|-------|----------|--------------|
| 2 | Observational | $\mathcal{TN}_{[-1.5,\,1.5]}(0,\,1)$ |
|   | Intervention 1 | $\mathcal{TN}_{[-1.5,\,1.5]}(-0.75,\,0.5)$ |
|   | Intervention 2 | $\mathcal{TN}_{[-1.5,\,1.5]}(\,0.75,\,0.5)$ |
| 4 | Observational | $\mathcal{TN}_{[-1.5,\,1.5]}(0,\,1)$ |
|   | Intervention 1 | $\mathcal{TN}_{[-1.5,\,1.5]}(-0.75,\,0.5)$ |
|   | Intervention 2 | $\mathcal{TN}_{[-1.5,\,1.5]}(\,0.75,\,0.5)$ |
| 6 | Observational | $\mathcal{TN}_{[0,\,3]}(1.5,\,1)$ |
|   | Intervention 1 | $\mathcal{TN}_{[0,\,3]}(2.25,\,0.5)$ |
|   | Intervention 2 | $\mathcal{TN}_{[0,\,3]}(\,0.75,\,0.5)$ |

Table 7: Sampling distributions for the in-distribution (ID) dataset. These truncated normal distributions define the observational and interventional data used for the two-camera Independent and Occlusion experiments.

| Joint | Scenario | Distribution |
|-------|----------|--------------|
| 1 | Observational | $\mathcal{TN}_{[0,\,3]}(1.2,\,0.4)$ |
|   | Intervention 1 | $\mathcal{TN}_{[0,\,3]}(2.0,\,0.4)$ |
|   | Intervention 2 | $\mathcal{TN}_{[0,\,3]}(\,0.6,\,0.4)$ |
| 2 | Observational | $\mathcal{TN}_{[-1.5,\,1.5]}(0,\,0.4)$ |
|   | Intervention 1 | $\mathcal{TN}_{[-1.5,\,1.5]}(0.7,\,0.4)$ |
|   | Intervention 2 | $\mathcal{TN}_{[-1.5,\,1.5]}(\,-0.7,\,0.4)$ |
| 3 | Observational | $\mathcal{TN}_{[-1.5,\,1.5]}(0,\,0.4)$ |
|   | Intervention 1 | $\mathcal{TN}_{[-1.5,\,1.5]}(0.7,\,0.4)$ |
|   | Intervention 2 | $\mathcal{TN}_{[-1.5,\,1.5]}(\,-0.7,\,0.4)$ |
| 4 | Observational | $\mathcal{TN}_{[-1.5,\,1.5]}(0,\,0.4)$ |
|   | Intervention 1 | $\mathcal{TN}_{[-1.5,\,1.5]}(0.9,\,0.4)$ |
|   | Intervention 2 | $\mathcal{TN}_{[-1.5,\,1.5]}(\,-0.9,\,0.4)$ |
| 5 | Observational | $\mathcal{TN}_{[-1.5,\,1.5]}(0,\,0.4)$ |
|   | Intervention 1 | $\mathcal{TN}_{[-1.5,\,1.5]}(0.9,\,0.4)$ |
|   | Intervention 2 | $\mathcal{TN}_{[-1.5,\,1.5]}(\,-0.9,\,0.4)$ |
| 6 | Observational | $\mathcal{TN}_{[0,\,3]}(1.5,\,0.4)$ |
|   | Intervention 1 | $\mathcal{TN}_{[0,\,3]}(2.4,\,0.4)$ |
|   | Intervention 2 | $\mathcal{TN}_{[0,\,3]}(\,0.7,\,0.4)$ |

Table 8: Observational sampling distributions for the out-of-distribution (OOD) dataset. These parameters define the OOD test sets for the Two-Camera Independent, Causal, and Occlusion experiments. The interventional distributions for the OOD dataset remain identical to those of the in-distribution dataset, as defined in Table 7.

| Joint | OOD Observational Distribution |
|-------|-------------------------------|
| 1 | $\mathcal{TN}_{[0,\,3]}(1.2,\,0.4)$ |
| 2 | $\mathcal{TN}_{[-1.5,\,1.5]}(0.0,\,0.4)$ |
| 3 | $\mathcal{TN}_{[-1.5,\,1.5]}(0.0,\,0.4)$ |
| 4 | $\mathcal{TN}_{[-1.5,\,1.5]}(0.8,\,0.4)$ |
| 5 | $\mathcal{TN}_{[-1.5,\,1.5]}(0.0,\,0.4)$ |
| 6 | $\mathcal{TN}_{[0,\,3]}(0.5,\,0.4)$ |

## E    RECONSTRUCTION FIGURES

Figures 8 and 9 provide a qualitative analysis of our pipeline's reconstruction performance. Specifically, Figures 8b and 8c compare an original image (Figure 8a) to its reconstructions from AE1 and AE2 respectively for the single-camera setup, while Figure 9 shows the equivalent comparison for the two-camera setup. We observe a slight degradation in the reconstruction quality of AE1 when trained on the two-camera dataset compared to the single-camera setup. As the final autoencoder, AE2, is trained on the feature maps from AE1, this reduction in quality of AE1 consequently affects the quality of the final AE2 reconstructions as well. We hypothesize that this performance difference is attributable to the increased data complexity of the multi-view dataset. The inclusion of multiple viewpoints introduces greater visual variance, presenting a more challenging reconstruction task for the autoencoder compared to the more constrained single-view data.

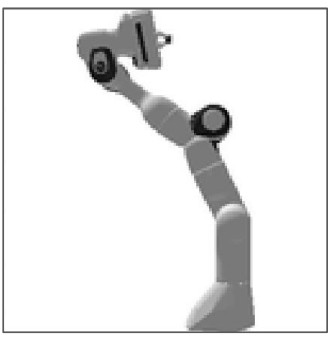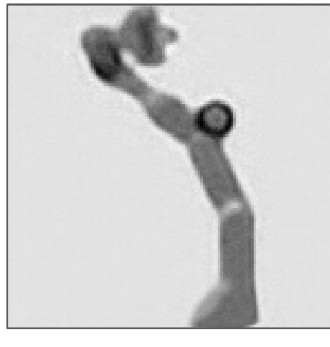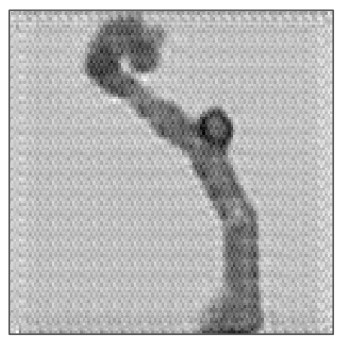

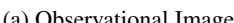

(a) Observational Image              (b) AE1 Reconstruction              (c) AE2 Reconstruction

Figure 8: Visual comparison of the reconstruction quality at each stage of our pipeline for the single camera setup. (a) The original input image. (b) The reconstruction from the first autoencoder (AE1). (c) The final reconstruction from the second autoencoder (AE2)

## F    CAUSAL DATASET GENERATION

The observational data is generated from a linear structural equation model (SEM) depicted in Figure 10. Specifically, we first sample the root node joints $J_1, J_2, J_4$ from the relevant distributions in Table 9. Then, values of $\{J_1, \ldots, J_6\}$ are determined by linear functions of their ancestors using the weights provided in Figure 10 plus noise $(\epsilon_1, \ldots, \epsilon_6)$ sampled from $\mathcal{TN}_{[0,\,1]}(0.0,\,0.1)$ distribution.

$$J_1 = J_1 + \epsilon_1 \qquad J_3 = 0.88J_2 + \epsilon_3$$
$$J_2 = J_2 + \epsilon_2 \qquad J_5 = 0.26J_3 + \epsilon_5$$
$$J_4 = J_4 + \epsilon_4 \qquad J_6 = 0.24J_1 + 0.31J_2 + 0.37J_3 + 0.15J_5 + \epsilon_6$$

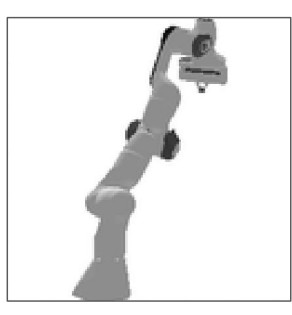 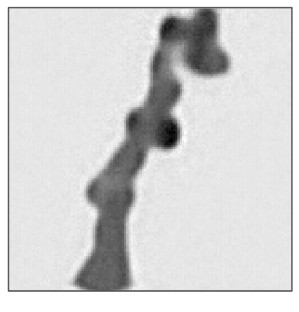 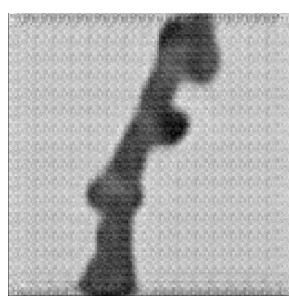

    (a) Observational Image        (b) AE1 Reconstruction        (c) AE2 Reconstruction

Figure 9: Visual comparison of the reconstruction quality at each stage of our pipeline using the two camera independent model. (a) The original input image. (b) The reconstruction from the first autoencoder (AE1). (c) The final reconstruction from the second autoencoder (AE2)

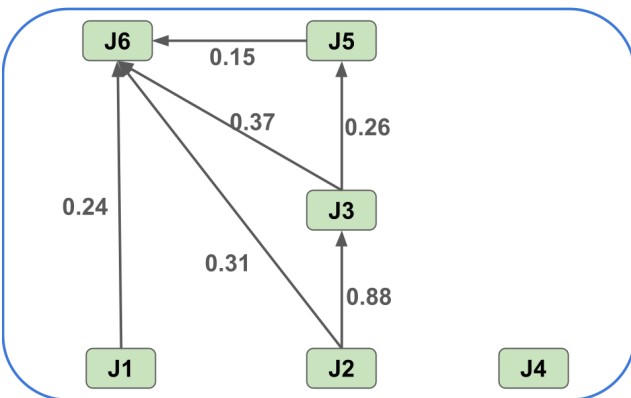

Figure 10: The causal model of the robot joints used to generate the dataset. In this graph, J$i$ represents the angle of joint $i$.

An intervention on a variable $J_k$ is denoted by setting it to a new value $J'_k$ sampled from the interventional distribution of choice in Table 9 — this works because we consider stochastic hard interventions that sets the interventional values directly equal to the exogenous noise variables. Next, we provide the equations that describe the system's behavior for interventions on each joint.

**Intervention on joint 1**    Under $do(J_1 := J'_1)$: The equation for $J_6$ depends on the new value $J'_1$ and others being unaffected and $J_1$ takes the value $J'_1$.

$$J_6 = 0.24J'_1 + 0.31J_2 + 0.37J_3 + 0.15J_5$$

**Intervention on joint 2**    Under $do(J_2 := J'_2)$: The equations for the descendants of $J_2$ are updated as given below.

$$J_3 = 0.88J'_2 + \epsilon_3$$
$$J_5 = 0.26J_3 + \epsilon_5$$
$$J_6 = 0.24J_1 + 0.31J'_2 + 0.37J_3 + 0.15J_5 + \epsilon_6$$

**Intervention on joint 3**    Under $do(J_3 := J'_3)$: The descendants of $J_3$ are affected.

$$J_5 = 0.26J'_3 + \epsilon_5$$
$$J_6 = 0.24J_1 + 0.31J_2 + 0.37J'_3 + 0.15J_5 + \epsilon_6$$

**Intervention on joint 4**    Under $do(J_4 := J'_4)$: No downstream variables are affected by an intervention on $J_4$, suggesting it is a root node with no children in this system.

Table 9: Sampling distributions for observational and interventional settings for causal dataset corresponding to the causal graph Figure 10

| Joint | Scenario | Distribution |
|---|---|---|
| 1 | Observational | $\mathcal{TN}_{[0,\,3]}(1.2,\ 0.4)$ |
| | Intervention 1 | $\mathcal{TN}_{[0,\,3]}(2.0,\ 0.4)$ |
| | Intervention 2 | $\mathcal{TN}_{[0,\,3]}(0.6,\ 0.4)$ |
| 2 | Observational | $\mathcal{TN}_{[-1.5,\,1.5]}(0,\ 0.4)$ |
| | Intervention 1 | $\mathcal{TN}_{[-1.5,\,1.5]}(0.7,\ 0.4)$ |
| | Intervention 2 | $\mathcal{TN}_{[-1.5,\,1.5]}(-0.7,\ 0.4)$ |
| 3 | Observational | Not a root node |
| | Intervention 1 | $\mathcal{TN}_{[-1.5,\,1.5]}(0.7,\ 0.4)$ |
| | Intervention 2 | $\mathcal{TN}_{[-1.5,\,1.5]}(-0.7,\ 0.4)$ |
| 4 | Observational | $\mathcal{TN}_{[-1.5,\,1.5]}(0,\ 0.4)$ |
| | Intervention 1 | $\mathcal{TN}_{[-1.5,\,1.5]}(0.9,\ 0.4)$ |
| | Intervention 2 | $\mathcal{TN}_{[-1.5,\,1.5]}(-0.9,\ 0.4)$ |
| 5 | Observational | Not a root node |
| | Intervention 1 | $\mathcal{TN}_{[-1.5,\,1.5]}(0.9,\ 0.4)$ |
| | Intervention 2 | $\mathcal{TN}_{[-1.5,\,1.5]}(-0.9,\ 0.4)$ |
| 6 | Observational | Not a root node |
| | Intervention 1 | $\mathcal{TN}_{[0,\,3]}(2.4,\ 0.4)$ |
| | Intervention 2 | $\mathcal{TN}_{[0,\,3]}(0.7,\ 0.4)$ |

**Intervention on joint 5**  Under $do(J_5 := J_5')$: The equation for $J_6$ is updated.

$$J_6 = 0.24 J_1 + 0.31 J_2 + 0.37 J_3 + 0.15 J_5' + \epsilon_6$$

**Intervention on joint 6**  Under $do(J_6 := J_6')$: No changes occur in any other variables except $J_6$, as $J_6$ is a sink node (it does not cause other variables in the system).

# G  SCATTER PLOTS

To provide a qualitative assessment of our models' disentanglement capabilities, we visualize the relationship between the learned latent variables and their corresponding ground-truth joint angles. Figures 11 and 12 present these scatter plots for the two-camera Independent and Causal models, respectively. Each plot is generated from the single best trial (out of 15) for that specific joint, determined by the highest MCC score. This visualization of the best-case performance complements the aggregated statistics reported in our main results tables.

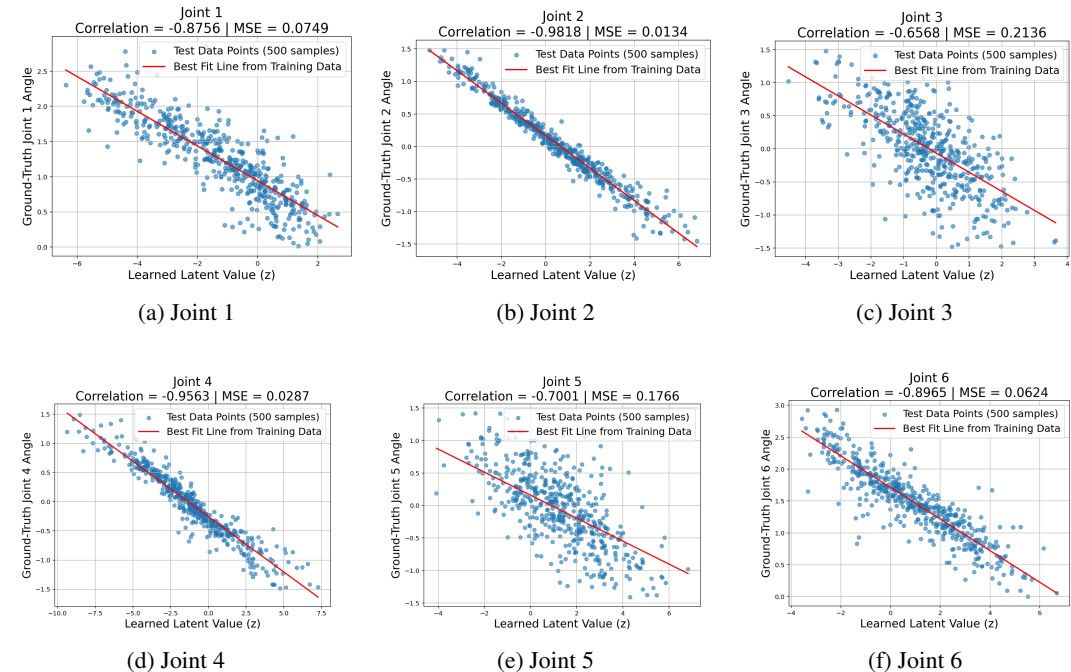

Figure 11: Scatter plots evaluating the two-camera Independent model on the in-distribution (ID) test set (Table7) . Each plot visualizes the relationship between a learned latent variable and its corresponding ground-truth joint angle. The displayed Correlation and MSE values correspond to the single best trial out of 15 runs, while the results presented in Table 15, Table 2 and Table 12 correspond to the mean statistics.

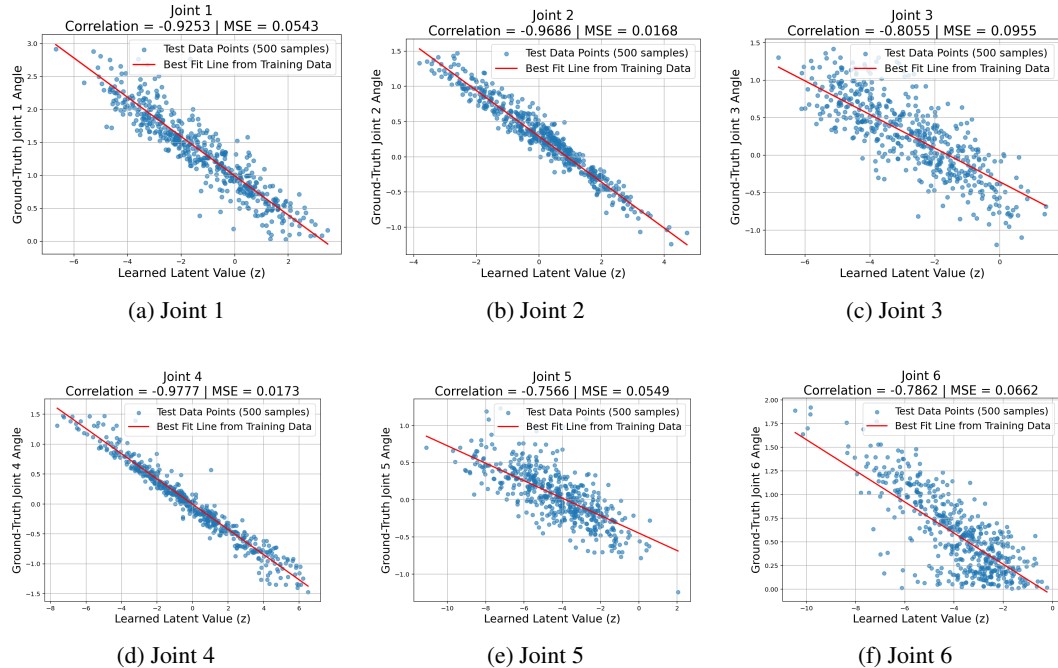

Figure 12: Scatter plots evaluating the two-camera causal model on the causally-generated test set. Each plot visualizes the relationship between a learned latent variable and its corresponding ground-truth joint angle. The displayed Correlation and MSE values correspond to the single best trial out of 15 runs, while the results presented in Table 15, Table 2 and Table 12 correspond to the mean statistics.

# H  ROBOPEPP

We pre-train a ViT-Huge model using the I-JEPA objective on 128x128x3 images. The core masking strategy involves one large context block (85-100% scale) and four smaller, non-overlapping target blocks (15-20% scale). The model is trained for 10 epochs with a batch size of 32, using a cosine learning rate schedule with a peak learning rate of $1e-3$. The target encoder's weights are updated with an exponential moving average (EMA). Notably, we only use random resized cropping for data augmentation, disabling strong augmentations like color jitter and horizontal flipping. Training is accelerated with bfloat16 mixed-precision.Our training is a two-stage process. In the first stage, we pre-train a ViT-Huge backbone using the I-JEPA self-supervised objective on the full ROPES dataset, comprising 260k images, as described previously. For the second stage, we fine-tune our JointNet model for a robotic perception task on the Panda robot. This supervised training uses the smaller RoboPEPP dataset (1%, 5%, 10%, 100%) with a batch size of 8. The model is trained for (100, 75, 25, 15) epochs respectively using a one-cycle learning rate schedule with a maximum learning rate of $1e-4$ and a weight decay of $1e-5$. Table 10, Table 11,Table 12, Table 13, Table 14 show a detailed analysis of the ROPES and RoboPEPP methodology.

Table 10: Per-joint Mean SquaredError (MSE) for the RoboPEPP model in radians squared ($rad^2$). The table presents an ablation study on the effect of varying training data labels, evaluated on both in-distribution (ID) test set (Table7) and out-of-distribution (OOD) test set (Table8).

| Dataset | Distribution | Epochs | Patch Size | Joint 1 | Joint 2 | Joint 3 | Joint 4 | Joint 5 | Joint 6 |
|---------|-------------|--------|-----------|---------|---------|---------|---------|---------|---------|
| 2.6k | ID | 100 | - | 0.136 | 0.039 | 0.237 | 0.053 | 0.253 | 0.200 |
| 2.6k | ID | 100 | 16 | 0.166 | 0.030 | 0.287 | 0.072 | 0.309 | 0.218 |
| 2.6k | ID | 100 | 32 | 0.186 | 0.060 | 0.305 | 0.125 | 0.331 | 0.263 |
| 2.6k | ID | 100 | 64 | 0.412 | 0.358 | 0.462 | 1.328 | 0.515 | 0.573 |
| 2.6k | OOD | 100 | - | 0.202 | 0.054 | 0.374 | 0.101 | 0.378 | 0.454 |
| 2.6k | OOD | 100 | 16 | 0.220 | 0.058 | 0.413 | 0.117 | 0.412 | 0.432 |
| 2.6k | OOD | 100 | 32 | 0.235 | 0.085 | 0.480 | 0.164 | 0.346 | 0.461 |
| 2.6k | OOD | 100 | 64 | 0.496 | 0.334 | 0.538 | 1.156 | 0.529 | 0.570 |
| 13k | ID | 75 | - | 0.075 | 0.017 | 0.134 | 0.024 | 0.153 | 0.081 |
| 13k | ID | 75 | 16 | 0.081 | 0.021 | 0.155 | 0.040 | 0.189 | 0.111 |
| 13k | ID | 75 | 32 | 0.140 | 0.050 | 0.304 | 0.198 | 0.299 | 0.232 |
| 13k | ID | 75 | 64 | 0.446 | 0.474 | 0.648 | 2.088 | 0.976 | 0.501 |
| 13k | OOD | 75 | - | 0.083 | 0.023 | 0.202 | 0.063 | 0.275 | 0.225 |
| 13k | OOD | 75 | 16 | 0.127 | 0.040 | 0.304 | 0.088 | 0.321 | 0.388 |
| 13k | OOD | 75 | 32 | 0.162 | 0.058 | 0.445 | 0.299 | 0.582 | 0.718 |
| 13k | OOD | 75 | 64 | 0.489 | 0.412 | 0.489 | 1.926 | 1.348 | 1.255 |
| 26k | ID | 25 | - | 0.030 | 0.010 | 0.072 | 0.022 | 0.091 | 0.063 |
| 26k | ID | 25 | 16 | 0.034 | 0.011 | 0.071 | 0.037 | 0.093 | 0.073 |
| 26k | ID | 25 | 32 | 0.077 | 0.036 | 0.194 | 0.180 | 0.181 | 0.136 |
| 26k | ID | 25 | 64 | 0.375 | 0.380 | 0.529 | 2.808 | 0.524 | 1.474 |
| 26k | OOD | 25 | - | 0.064 | 0.020 | 0.125 | 0.037 | 0.173 | 0.152 |
| 26k | OOD | 25 | 16 | 0.050 | 0.019 | 0.126 | 0.055 | 0.197 | 0.146 |
| 26k | OOD | 25 | 32 | 0.101 | 0.066 | 0.230 | 0.262 | 0.308 | 0.202 |
| 26k | OOD | 25 | 64 | 0.394 | 0.297 | 0.497 | 2.475 | 0.582 | 1.028 |
| 260k | ID | 15 | - | 0.003 | 0.001 | 0.007 | 0.003 | 0.010 | 0.011 |
| 260k | ID | 15 | 16 | 0.016 | 0.004 | 0.026 | 0.007 | 0.030 | 0.011 |
| 260k | ID | 15 | 32 | 0.066 | 0.036 | 0.097 | 0.053 | 0.090 | 0.045 |
| 260k | ID | 15 | 64 | 0.268 | 0.517 | 0.593 | 1.891 | 0.478 | 0.430 |
| 260k | OOD | 15 | - | 0.007 | 0.003 | 0.013 | 0.004 | 0.018 | 0.019 |
| 260k | OOD | 15 | 16 | 0.029 | 0.008 | 0.059 | 0.014 | 0.081 | 0.028 |
| 260k | OOD | 15 | 32 | 0.130 | 0.060 | 0.210 | 0.088 | 0.201 | 0.125 |
| 260k | OOD | 15 | 64 | 0.262 | 0.447 | 0.649 | 1.437 | 0.465 | 0.616 |

# I ROPES

Table 11: Mean Squared Error (MSE) in radians squared ($rad^2$) for each joint under various experimental conditions for two camera angles. The table compares performance on in-distribution (ID) test set (Table7) and out-of-distribution (OOD) test set (Table8) for independent, causal, and occluded inference models.

| Experiment | Distribution | Patch Size | Joint 1 | Joint 2 | Joint 3 | Joint 4 | Joint 5 | Joint 6 |
|---|---|---|---|---|---|---|---|---|
| Independent | ID | – | 0.083 | 0.015 | 0.217 | 0.035 | 0.198 | 0.080 |
| Causal | ID | – | 0.058 | 0.020 | 0.106 | 0.019 | 0.051 | 0.070 |
| Occlusion | ID | 16 | 0.089 | 0.018 | 0.225 | 0.044 | 0.200 | 0.082 |
| Occlusion | ID | 32 | 0.101 | 0.025 | 0.245 | 0.082 | 0.225 | 0.145 |
| Occlusion | ID | 64 | 0.186 | 0.077 | 0.322 | 0.258 | 0.298 | 0.273 |
| Independent | OOD | – | 0.084 | 0.017 | 0.239 | 0.048 | 0.199 | 0.120 |
| Causal | OOD | – | 0.108 | 0.044 | 0.241 | 0.085 | 0.219 | 0.116 |
| Occlusion | OOD | 16 | 0.092 | 0.019 | 0.249 | 0.054 | 0.205 | 0.109 |
| Occlusion | OOD | 32 | 0.103 | 0.024 | 0.288 | 0.079 | 0.226 | 0.140 |
| Occlusion | OOD | 64 | 0.201 | 0.087 | 0.356 | 0.227 | 0.331 | 0.216 |

Table 12: Comparison of MCC and MSE($rad^2$) for each joint across two model settings with error bars. Mean and Std Dev are calculated across the 15 runs as discussed in the section 4. Values are reported as Mean ± Std Dev.

| | 2C, indep. | | 2C, causal | |
|---|---|---|---|---|
| Joint Angle | MCC | MSE | MCC | MSE |
| Joint 1 | $0.874 \pm 0.004$ | $0.083 \pm 0.006$ | $0.921 \pm 0.003$ | $0.058 \pm 0.003$ |
| Joint 2 | $0.979 \pm 0.001$ | $0.015 \pm 0.001$ | $0.966 \pm 0.002$ | $0.020 \pm 0.001$ |
| Joint 3 | $0.634 \pm 0.010$ | $0.217 \pm 0.011$ | $0.788 \pm 0.003$ | $0.106 \pm 0.005$ |
| Joint 4 | $0.950 \pm 0.002$ | $0.035 \pm 0.002$ | $0.976 \pm 0.001$ | $0.019 \pm 0.001$ |
| Joint 5 | $0.679 \pm 0.010$ | $0.198 \pm 0.011$ | $0.742 \pm 0.006$ | $0.051 \pm 0.003$ |
| Joint 6 | $0.884 \pm 0.005$ | $0.080 \pm 0.007$ | $0.756 \pm 0.010$ | $0.070 \pm 0.003$ |

2C = two cameras; indep. and causal refer to the joint distribution model.

## J  COMPARISON BETWEEN ROPES AND RoboPEPP

Table 13: Comparison of In-Distribution (ID) Mean Squared Error (MSE) in radians squared ($rad^2$) for the ROPES and RoboPEPP models. Results are shown per joint under various experimental conditions.

| Model | Experiment | Patch Size | Joint 1 | Joint 2 | Joint 3 | Joint 4 | Joint 5 | Joint 6 |
|---|---|---|---|---|---|---|---|---|
| ROPES | Independent | – | 0.083 | 0.015 | 0.217 | 0.035 | 0.198 | 0.080 |
| | Causal | – | 0.058 | 0.020 | 0.106 | 0.019 | 0.051 | 0.070 |
| | Occlusion | 16 | 0.089 | 0.018 | 0.225 | 0.044 | 0.200 | 0.082 |
| | Occlusion | 32 | 0.101 | 0.025 | 0.245 | 0.082 | 0.225 | 0.145 |
| | Occlusion | 64 | 0.186 | 0.077 | 0.322 | 0.258 | 0.298 | 0.273 |
| RoboPEPP | 2.6k Dataset | – | 0.136 | 0.039 | 0.237 | 0.053 | 0.253 | 0.200 |
| | 2.6k Dataset | 16 | 0.166 | 0.030 | 0.287 | 0.072 | 0.309 | 0.218 |
| | 2.6k Dataset | 32 | 0.186 | 0.060 | 0.305 | 0.125 | 0.331 | 0.263 |
| | 2.6k Dataset | 64 | 0.412 | 0.358 | 0.462 | 1.328 | 0.515 | 0.573 |
| | 13k Dataset | – | 0.075 | 0.017 | 0.134 | 0.024 | 0.153 | 0.081 |
| | 13k Dataset | 16 | 0.081 | 0.021 | 0.155 | 0.040 | 0.189 | 0.111 |
| | 13k Dataset | 32 | 0.140 | 0.050 | 0.304 | 0.198 | 0.299 | 0.232 |
| | 13k Dataset | 64 | 0.446 | 0.474 | 0.648 | 2.088 | 0.976 | 0.501 |
| | 26k Dataset | – | 0.030 | 0.010 | 0.072 | 0.022 | 0.091 | 0.063 |
| | 26k Dataset | 16 | 0.034 | 0.011 | 0.071 | 0.037 | 0.093 | 0.073 |
| | 26k Dataset | 32 | 0.077 | 0.036 | 0.194 | 0.180 | 0.181 | 0.136 |
| | 26k Dataset | 64 | 0.375 | 0.380 | 0.529 | 2.808 | 0.524 | 1.474 |
| | 260k Dataset | – | 0.003 | 0.001 | 0.007 | 0.003 | 0.010 | 0.011 |
| | 260k Dataset | 16 | 0.016 | 0.004 | 0.026 | 0.007 | 0.030 | 0.011 |
| | 260k Dataset | 32 | 0.066 | 0.036 | 0.097 | 0.053 | 0.090 | 0.045 |
| | 260k Dataset | 64 | 0.268 | 0.517 | 0.593 | 1.891 | 0.478 | 0.430 |

Table 14: Comparison of Out-of-Distribution (OOD) Mean Squared Error (MSE) in radians squared (rad$^2$) for the ROPES and RoboPEPP models. Results are shown per joint under various experimental conditions.

| Model | Experiment | Patch Size | Joint 1 | Joint 2 | Joint 3 | Joint 4 | Joint 5 | Joint 6 |
|---|---|---|---|---|---|---|---|---|
| ROPES | Independent | – | 0.084 | 0.017 | 0.239 | 0.048 | 0.199 | 0.120 |
| | Causal | – | 0.108 | 0.044 | 0.241 | 0.085 | 0.219 | 0.116 |
| | Occlusion | 16 | 0.092 | 0.019 | 0.249 | 0.054 | 0.205 | 0.109 |
| | Occlusion | 32 | 0.103 | 0.024 | 0.288 | 0.079 | 0.226 | 0.140 |
| | Occlusion | 64 | 0.201 | 0.087 | 0.356 | 0.227 | 0.331 | 0.216 |
| RoboPEPP | 2.6k Dataset | – | 0.202 | 0.054 | 0.374 | 0.101 | 0.378 | 0.454 |
| | 2.6k Dataset | 16 | 0.220 | 0.058 | 0.413 | 0.117 | 0.412 | 0.432 |
| | 2.6k Dataset | 32 | 0.235 | 0.085 | 0.480 | 0.164 | 0.346 | 0.461 |
| | 2.6k Dataset | 64 | 0.496 | 0.334 | 0.538 | 1.156 | 0.529 | 0.570 |
| | 13k Dataset | – | 0.083 | 0.023 | 0.202 | 0.063 | 0.275 | 0.225 |
| | 13k Dataset | 16 | 0.127 | 0.040 | 0.304 | 0.088 | 0.321 | 0.388 |
| | 13k Dataset | 32 | 0.162 | 0.058 | 0.445 | 0.299 | 0.582 | 0.718 |
| | 13k Dataset | 64 | 0.489 | 0.412 | 0.489 | 1.926 | 1.348 | 1.255 |
| | 26k Dataset | – | 0.064 | 0.020 | 0.125 | 0.037 | 0.173 | 0.152 |
| | 26k Dataset | 16 | 0.050 | 0.019 | 0.126 | 0.055 | 0.197 | 0.146 |
| | 26k Dataset | 32 | 0.101 | 0.066 | 0.230 | 0.262 | 0.308 | 0.202 |
| | 26k Dataset | 64 | 0.394 | 0.297 | 0.497 | 2.475 | 0.582 | 1.028 |
| | 260k Dataset | – | 0.007 | 0.003 | 0.013 | 0.004 | 0.018 | 0.019 |
| | 260k Dataset | 16 | 0.029 | 0.008 | 0.059 | 0.014 | 0.081 | 0.028 |
| | 260k Dataset | 32 | 0.130 | 0.060 | 0.210 | 0.088 | 0.201 | 0.125 |
| | 260k Dataset | 64 | 0.262 | 0.447 | 0.649 | 1.437 | 0.465 | 0.616 |

# K    ABLATIONS

We conducted a series of experiments to find the proper training setting for ROPES model. Specifically, we investigated the effect of the learning rate, batch size, and weights of the loss components ($\lambda = 3$ in Eqn. (3)) on the model's ability to disentangle the latent features, as measured by the MCC score. All experiments in this section were performed using Autoencoder-2 with a 2-camera setup on the independent dataset for joint 4.

**Learning Rate.**    First, we analyzed the impact of the learning rate.    Figure 13 shows the training progress for different learning rates while keeping the batch size fixed at 128 and the weight of the sparsity loss constant ($\lambda = 3$ in Eqn. (3)). We observe that smaller learning rates, such as $5e - 5$ and $8e - 5$, lead to better performance.

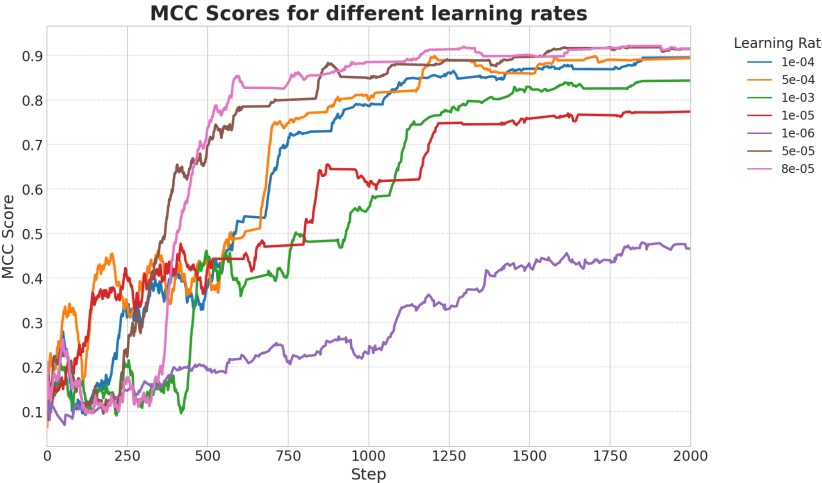

Figure 13: Evolution of MCC scores during training for Autoencoder-2 using a 2-camera setup on the independent dataset. The plot compares performance across various learning rates. Training is performed with a batch size of 128, and sparsity loss weight of $\lambda = 3$.

**Batch Size.** Next, we performed an ablation study on the batch size (Figure 14), using the optimal learning rate $5e - 5$ found previously. We observed that models with larger batch sizes converged faster; however, their final asymptotic performances were nearly identical, irrespective of the batch size.

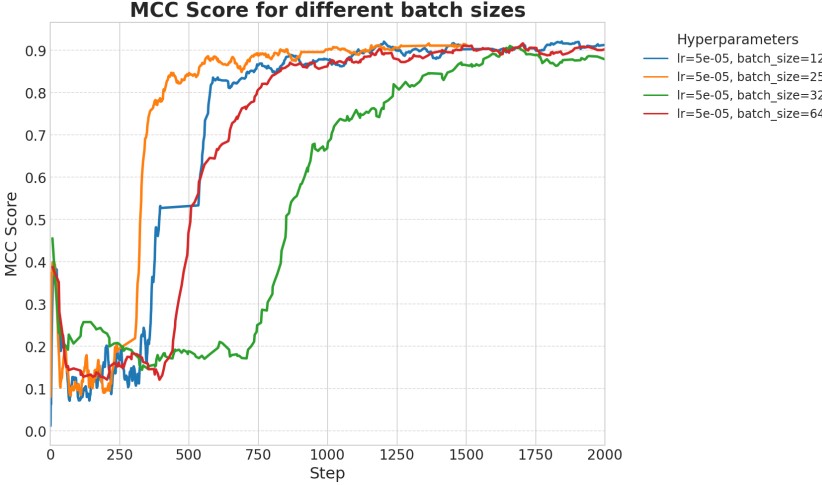

Figure 14: Evolution of MCC scores during training for Autoencoder-2 using a 2-camera setup on the independent dataset. The plot compares performance across various batch sizes. Training is performed with a learning rate of $5e - 5$ and sparsity loss weight of $\lambda = 3$.

**Loss Reweighting.** We also looked at how to balance the two training objectives: reconstructing the image and the sparsity loss. Figure 15 compares the MCC scores when we vary the weights given to the reconstruction loss versus the sparsity loss. For these experiments, we fixed the learning rate at $5e-5$ and batch size at 128. The figure shows that giving a significant weight to the score-based loss (i.e., sparsity loss) is crucial for disentanglement. This aligns with our theory that a reconstruction loss alone cannot perform disentanglement (e.g., MCC of less than 0.15 in the figure), but the score-based loss is the main driver of the disentanglement (e.g., over 0.9 MCC in the figure).

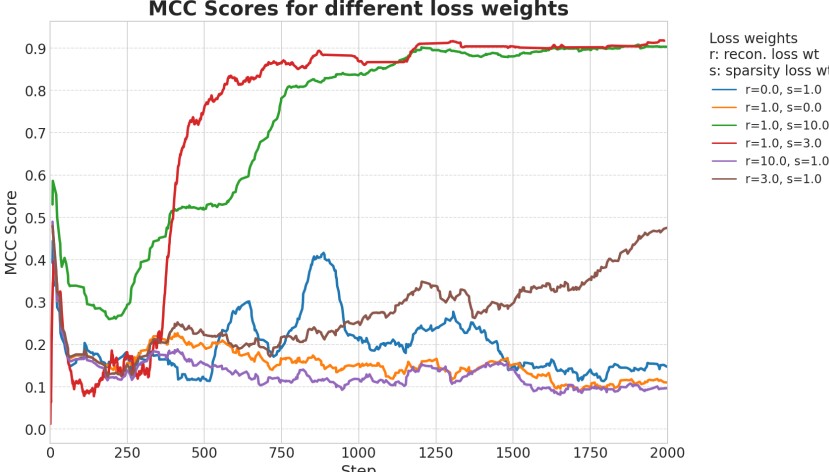

Figure 15: Evolution of MCC scores during training for Autoencoder-2 using a 2-camera setup on the independent dataset. The plot compares performance across various combinations of reconstruction loss weights and sparsity loss weights. Training is performed with a batch size of 128 and a fixed learning rate of $5e - 5$.

**Architectural Changes - Residual Connection.** The inclusion of intra-module residual connections—applied independently within the encoder and decoder architectures—is a key factor for optimal performance. This architectural choice as seen in the Figure 16 improves the reconstruction loss and hence the final MCC score.

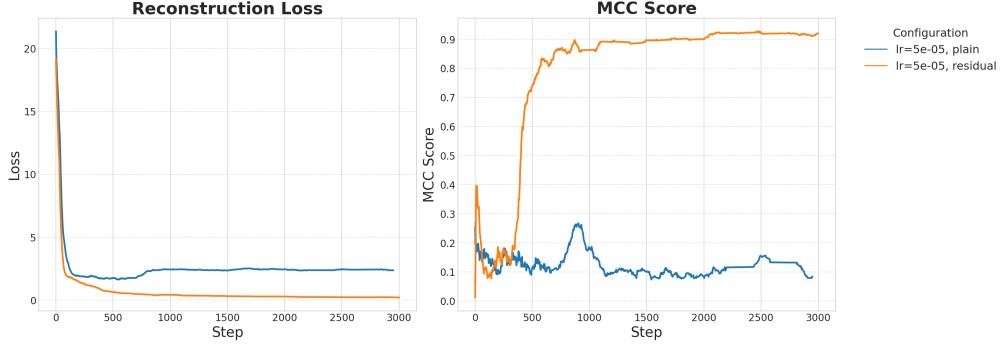

Figure 16: Ablation showing the advantage of having residual connections for better reconstruction loss and MCC score

**Calibration Samples.** As demonstrated in Table 15, the model's Mean Squared Error (MSE) remains stable even when the number of calibration samples is reduced from 1000 to 100.

Table 15: MCC and MSE of ROPES across different Calibration settings. MSE is reported in radians squared.

| Calibration Samples | Joint 1 MCC | Joint 1 MSE | Joint 2 MCC | Joint 2 MSE | Joint 3 MCC | Joint 3 MSE | Joint 4 MCC | Joint 4 MSE | Joint 5 MCC | Joint 5 MSE | Joint 6 MCC | Joint 6 MSE |
|---|---|---|---|---|---|---|---|---|---|---|---|---|
| 1000 samples | 0.874 | 0.083 | 0.979 | 0.015 | 0.634 | 0.217 | 0.950 | 0.035 | 0.679 | 0.198 | 0.884 | 0.080 |
| 500 samples | 0.863 | 0.094 | 0.979 | 0.015 | 0.618 | 0.234 | 0.945 | 0.037 | 0.677 | 0.191 | 0.895 | 0.075 |
| 100 samples | 0.881 | 0.079 | 0.975 | 0.019 | 0.612 | 0.227 | 0.942 | 0.039 | 0.661 | 0.199 | 0.879 | 0.084 |

**Lighting Condition.** Table 16 evaluates the model's performance under distinct lighting scenarios, simulated by scaling pixel brightness to 0.8x (dark) and 1.25x (light) of the original. Performance degradation is most pronounced in the dark, particularly for Joint 5 and Joint 6, which exhibit a significant drop in MCC. In contrast, the remaining joints maintain strong performance with only minimal degradation under both conditions.

Table 16: MCC and MSE of ROPES across different lighting condition. MSE is reported in radians squared.

| Lighting Condition | Joint 1 MCC | Joint 1 MSE | Joint 2 MCC | Joint 2 MSE | Joint 3 MCC | Joint 3 MSE | Joint 4 MCC | Joint 4 MSE | Joint 5 MCC | Joint 5 MSE | Joint 6 MCC | Joint 6 MSE |
|---|---|---|---|---|---|---|---|---|---|---|---|---|
| original | 0.874 | 0.083 | 0.979 | 0.015 | 0.634 | 0.217 | 0.950 | 0.035 | 0.679 | 0.198 | 0.884 | 0.080 |
| bright | 0.874 | 0.072 | 0.978 | 0.015 | 0.640 | 0.207 | 0.950 | 0.033 | 0.677 | 0.180 | 0.874 | 0.094 |
| dark | 0.811 | 0.107 | 0.940 | 0.046 | 0.520 | 0.211 | 0.860 | 0.093 | 0.199 | 0.303 | 0.526 | 0.265 |

## L  ROBOSUITE EXPERIMENTS

To validate that our method, ROPES, is simulator-agnostic, we tested its performance in the more realistic `robosuite` environment Zhu et al. (2020). This simulator provides a robotics setup where the task was to perform the stacking of cubes, allowing us to test our model in a setting with more complex physics and visuals.

**Experimental Setup.**  The experimental design mirrored our previous tests to ensure a fair comparison. The dataset had 260k samples with the same in-distribution (ID) setup described in Table 7. The images were scaled up from $128 \times 128$ to $512 \times 512$ to capture the surrounding details with a higher resolution. The camera configuration also remained the same, with two cameras positioned perpendicularly to each other at $45°$ and $135°$. For training the network components, we used a learning rate of $1 \times 10^{-4}$ for autoencoder-1, the LDR, and autoencoder-2. The reweighting factor $\lambda$ for the autoencoder-2 loss function was set to 10, as defined in Equation 3.

**Results and Analysis.**  Despite the increased complexity of the `robosuite` environment leading to a slightly higher reconstruction loss, our ROPES method demonstrated decent performance. For the in-distribution test samples, the model was able to accurately recover the primary joint states. We measured MCC of 0.90 for joint 1. The result shows promise that ROPES can generalize to different simulators.

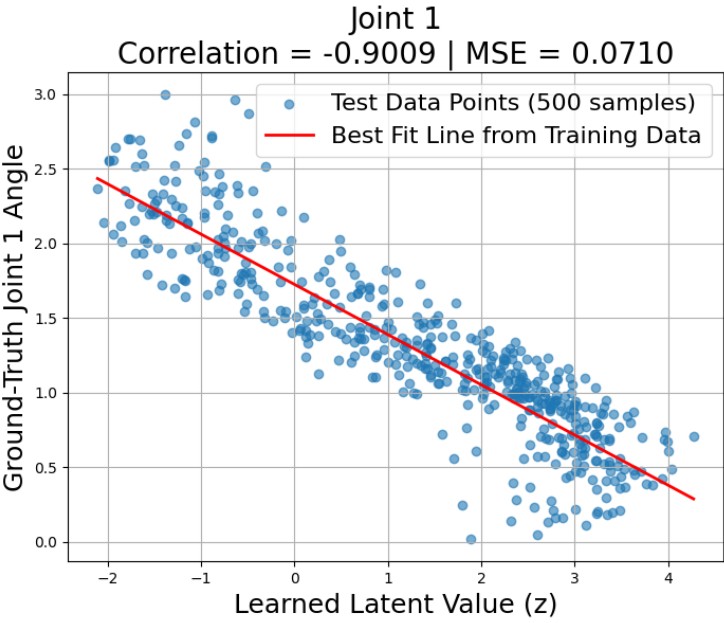

Figure 17: Scatter plot evaluating the two-camera Independent model on the in-distribution (ID) test set (Table7) . Each plot visualizes the relationship between a learned latent variable and its corresponding ground-truth joint angle.

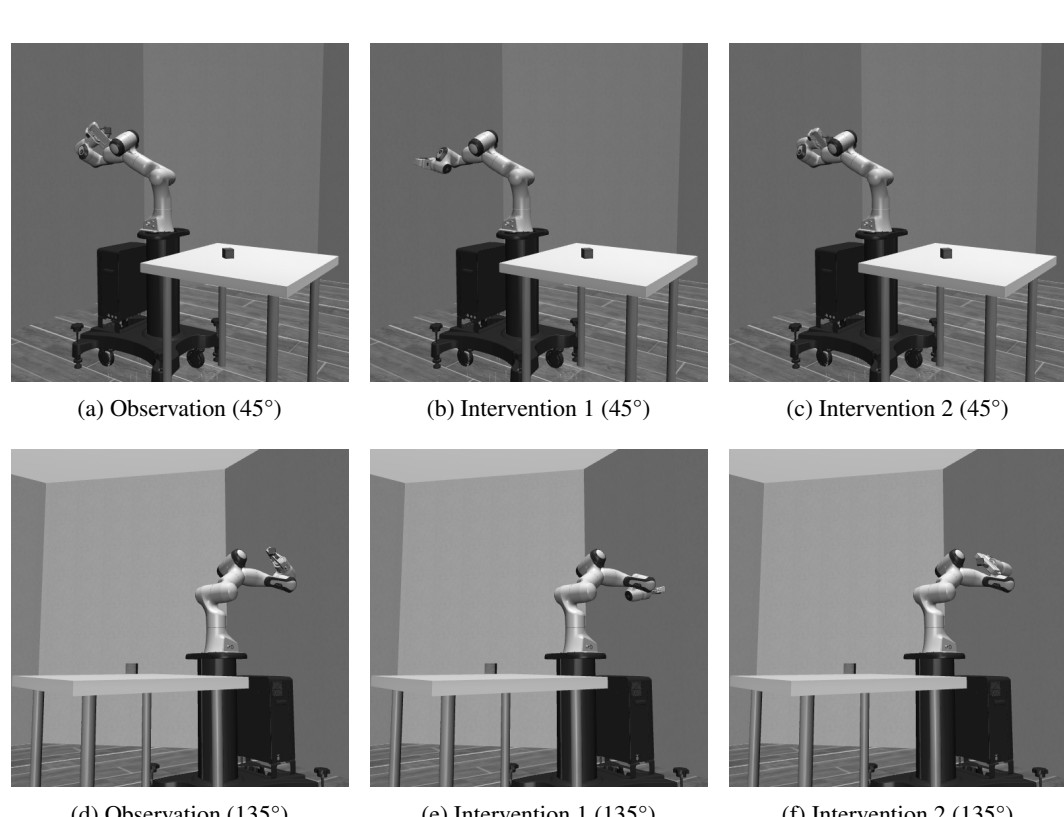

(a) Observation (45°)     (b) Intervention 1 (45°)     (c) Intervention 2 (45°)

(d) Observation (135°)     (e) Intervention 1 (135°)     (f) Intervention 2 (135°)

Figure 18: Two different hard interventions, each shown from two camera angles.

# M   LARGE LANGUAGE MODEL USAGE

In adherence to the ICLR 2026 policy, we state that the human authors entirely made the scientific contributions and wrote the core text of the paper. We have used large language models only as a writing assistance tool for tasks such as grammar check and rephrasing sentences to improve clarity.

