# OpenReview forum: "ROPES: Robotic Pose Estimation via Score-based Causal Representation Learning"
_ICLR.cc/2026/Conference — Submitted to ICLR 2026_

### Official Review · Reviewer_Nc2F · 2025-10-26

**Soundness:** 3
**Presentation:** 3
**Contribution:** 2
**Rating:** 4
**Confidence:** 4

**Summary:**

This paper shows that we can learn a disentangled representation of robot joints from images without using joint angle labels. The key idea is to collect data where one joint is intentionally varied at a time; by looking at how the image distribution changes under each of these controlled moves, the model learns a latent space where each dimension corresponds to a specific joint. The contribution is not about precise pose estimation, but about demonstrating that robot joint factors can be separated and identified purely from visual interventions, without supervised joint annotations. Experiments on a simulated Panda arm confirm that each latent dimension strongly correlates with its corresponding joint angle, and a simple linear probe is sufficient to recover accurate joint values from the learned representation.

**Strengths:**

The paper is built on a solid theoretical foundation, using score-based causal representation learning to ensure that each latent variable can be meaningfully linked to a joint. It introduces a practical intervention strategy for robotics, where one joint is moved at a time so the model can learn which visual changes correspond to which joint in the representation. The experiments show clear disentanglement, with each latent dimension strongly matching one joint’s variation. Finally, the model also shows robustness to occlusion, meaning it can still identify the underlying joint representation even when part of the robot in the image is blocked.

**Weaknesses:**

The method still relies on knowing which joint is being varied during data collection, so while no joint angle labels are used, the learning is not fully unsupervised. The experiments are conducted entirely in simulation, leaving open how well the approach would transfer to real robot camera setups with more visual noise and complexity. The learned latent dimensions do not directly correspond to physical joint angles and require a small labelled calibration step to decode them into meaningful values. Finally, the work does not show any downstream usage of the learned representation, so it remains unclear how beneficial this disentanglement is for actual control or manipulation tasks. The paper would be significantly strengthened if these limitations were addressed through additional real-world experiments, calibration-free evaluations, and demonstrations on downstream control or manipulation tasks.

**Questions:**

Based on the weakness,
1. Since the method requires knowing which joint is being varied during data collection, how dependent is the approach on this joint-index supervision, and can it work without it?
2. All results are in simulation. Do you have any evidence that the method can generalize to real robot images with noise and less controlled viewpoints?
3. The latent needs a linear calibration step to recover joint angles. How stable is this mapping, and how much labelled data is needed to make it reliable?
4. The paper does not show downstream use. Can you demonstrate how the learned representation would actually benefit robot control, planning, or imitation tasks?

---

> ### Author Response · Authors · 2025-11-21
> **Reply to Reviewer Nc2F**
>
> **CRL versus robotics perspective:** Before addressing specific questions and comments, we would like to invite you to first check our general response regarding the intended scope of this paper and its key contributions from a CRL perspective. We believe that perspective is important for evaluating the contributions.
>
> **Contribution to CRL**: Following up on the notes in the general response, we would like to emphasize that while robotics has been a key motivating application for CRL, despite significant advances in CRL, there is still no study that establishes the relevance of CRL to a well-defined specific robotics application. Formalizing the robot pose estimation as a CRL problem is a significant contribution of our paper. Specifically, we demonstrate how to frame the well-defined pose estimation problem as a CRL task and show how to create diverse data aligned with theoretical necessities.
>
> **New experiments on Robosuite**: To address the comments on the need for more complex data, we identified Robosuite simulator which is based on the MuJoCo physics engine to be a challenging test bed to try out ideas. This simulator provides higher resolution images (512x512 versus the previous 128x128) with realistic-looking backgrounds. We have added additional experiments establishing successful disentanglement results (Figure 16 in Appendix L).  For instance, we have been able to disentangle Joint 1 with an MCC of 0.91, showing the promise of our method in even more scaled-up settings.
>
> **Linear calibration**: We recall that we report results in two metrics: (1) mean correlation coefficient (MCC) for raw CRL output, and (2) mean squared error (MSE) for CRL output post-calibration. The first metric (MCC) does not use any label and has been the standard evaluation measure in CRL. For the second metric (MSE), we kindly note that a simple calibration step is **inevitable** to align the scale of the learned representations. Otherwise, a direct MSE evaluation is not possible, as the “unit” of the target variable will be unspecified, e.g., inches vs. centimeters, and taking the difference between quantities with mismatching units would be meaningless. However, we have added Table 15 in Appendix K that suggests that we only need $100$ labeled samples to get the calibration right. We also do an ablation of the number of samples with calibration performance and we see that it quickly saturates.
>
> **Downstream usage**: Indeed, investigating CRL in the context of downstream tasks is the ultimate objective. We believe a necessary prerequisite step is to understand the problem at a more basic and fundamental level. However, as discussed in the general response, our scope is establishing such prerequisite steps. Pose estimation is a well-defined problem in robotics that is the basis for designing other tasks. In this paper, our objective is to show that pose estimation can be formalized and solved as a CRL problem. We believe that demonstrating how useful these pose estimates for subsequent problems (such as planning and state-based control) is beyond the scope of this paper and is an important future direction for the intersection of CRL and robotics communities.

---

### Official Review · Reviewer_xQzc · 2025-10-31

**Soundness:** 2
**Presentation:** 3
**Contribution:** 3
**Rating:** 4
**Confidence:** 4

**Summary:**

The paper utilises recent progress of score-based causal inference in the context of learning disentangled latent variables from robot observations. The central method leverages the techniques of revealing connection between latent and observable space dimensions through score difference, and proposes to use an established binary-classifier-based method to estimate the difference. The results are validation on learning disentangled latent variables corresponding to robot joints from image observations. Causal and independent joint models are set to a simulated franka robot under single and two-camera cases. The causality-based representation learning shows good data efficiency in the domain with sparse labels and image occlusions.

**Strengths:**

* Exploration of causality-based representation learning in robotics and general machine learning is highly needed.
* The method design reads reasonable and not hard to grasp.
* The results look promising under certain experiment conditions.

**Weaknesses:**

* The main technical methods are from existing causal learning literature, which might be fine from an application perspective but nonetheless compromises the originality of the paper.
* It is hard to tell whether the proposed causal learning is consistently better than baselines, especially when 100% labels are available as in Table 2. This questions the significance of the results from an empirical perspective.
* The occlusion experiment condition looks a bit artificial and not rooted from a practical context. This may again harm the prospect of applying the proposed methods in real tasks.
* The method seems difficult to scale to cases with a large number of latent factors. See questions below.
* Needing interventional data, which might not always be available in typical robot datasets.

**Questions:**

* Can the paper elaborate more on the results from Table 2? Why ROPES numbers are bolded even RoboPEPP with 100% labels has significantly lower MSE?
* How can we expect the method to scale to more or an unknown number of latent factors when we have to train a binary classifier for each pair of variables?
* Line 454 states "robustness to real-world corruptions" while the next line goes with "we introduce artificial occlusions in the form of 32x32 white pixels squares". How can such a corruption pattern be regarded as "real-world corruption"?

---

> ### Author Response · Authors · 2025-11-21
> **Reply to Reviewer xQzc**
>
> **CRL versus robotics perspective:** We would like to thank the reviewer for thoughtful evaluation and questions. Before addressing specific questions and comments, we would like to invite you to first check our general response regarding the intended scope of this paper and its key contributions from a CRL perspective. We believe that perspective is important for evaluating the contributions.
>
> **Advances made by the paper:**
>
> **Contribution to CRL**: Following up on the notes in the general response, we would like to emphasize that while robotics has been a key motivating application for CRL, despite significant advances in CRL, there is still no study that establishes the relevance of CRL to a well-defined specific robotics application. Formalizing the robot pose estimation as a CRL problem is a significant contribution of our paper. Specifically, we demonstrate how to frame the well-defined pose estimation problem as a CRL task and show how to create diverse data aligned with theoretical necessities
>
> **Comparisons**: Since it is critical, at the expense of being repetitive, we would like to note that this paper’s contribution is advancing the applications of CRL by formalizing that a well-defined robotics problem can be posed and solved as a CRL problem. Our primary objective is not establishing that we are advancing the state-of-the-art on pose estimation. After this clarification, we elaborate on the results in Table 2.
>
> **Elaboration of Table 2**. We revised Table 2 (MSE evaluation) and added the number of labeled samples used for each row. Note that ROPES uses only ~100 labeled samples only for the calibration step, whereas even 5% of RoboPEPP corresponds to 13k images across two camera views with 6.5k uniquely labeled examples (since two camera views share the same labels). We added Table 15 in the Appendix K which ablates the calibration performance against the number of labelled samples. We see that about 100 labeled examples are enough to reach the performance of RoboPEPP with 6.5k labeled examples. We note that if MCC is the metric used, we need $0$ labels.  As for bolding, we picked the level of supervision for RoboPEPP that gives comparable performance with ROPES and bolded numbers from both methods accordingly. We removed this styling in the revision to avoid any overstatement. We acknowledge that when 100% of the labels (260k samples across two camera views and 130k uniquely labeled images) are available, the supervised baseline achieves stronger results than ROPES.
>
> **Scalability of ROPES**: We note that the computational complexity grows **linearly** in the available interventions, and the method disentangles only the intervened variables. Furthermore, in problems such as pose estimation, the number of key parameters of interest is typically small, as a key motivation of CRL is learning the low-dimensional underlying structure.
>
> **Occlusions:** We acknowledge that stating the used 32x32 masks as “real-world corruptions” is an overstatement, and we fix it in the revised version. Our goal in this experiment was not to mimic all real-world occluders, but to use a controlled corruption to test robustness and reconstruction ability from the sparsity-based loss. We kindly note that this type of artificial occlusion was also used by the related work, e.g., RoboPEPP.
>
> **Robot Intervention data and CRL:** We agree with the reviewer that interventional data might not be readily available in some robot datasets. What we show is that, when the joints can undergo mechanism changes, CRL can disentangle the pose variables. Interventions are one form of creating mechanism change. More generally, datasets that are collected under similar conditions could be formulated as interventions on a shared model, enabling potential use of CRL in other robotics problems.

---

> > ### Comment · Reviewer_xQzc · 2025-11-26
> >
> > Thanks for the clarification on the domain that the binary classifiers are trained on. Now I understand it is trained per the same joint under a pair of intervened configurations. I also agree with the clarified scope that the paper should be more categorised as an application paper and not the progress on causal inference on its own. From this perspective, the validation tasks look a bit contrived by using vision to retrieve information that can be read from proprioceptive. I would recommend to look at tasks where state sensing is not very obvious, like extracting the configuration of a cloth manipulated by a robot. In all, I stick to my evaluation and original score, in that it is nice to see the efforts of applying CRL in robot learning while the authors are encouraged to explore more realistic and significant scenarios.

---

> > > ### Author Response · Authors · 2025-11-27
> > >
> > > We would like to thank the reviewer for the additional comment. We are glad that the point regarding the pose-image labeled data is clarified, and how we can achieve what the baseline method achieves without requiring such data.
> > >
> > > We certainly agree that there are other alternative problems to further investigate. We, respectively, disagree with the reviewer that pose estimation is a contrived problem. In addition to the rich literature on pose estimation (with development as recent as in premier venues in 2025), we would like to highlight two more applications that critically need pose estimation without using labeled data:
> > >
> > > 1) **Vision-language action (VLA) models:** These models require labeling robot images available in the wild without corresponding pose-label information. Addressing this need requires exploiting the natural variations in robots data to completely disentangle individual variables. Our paper shows CRL can do this with variations introduced by actuations on the joints.
> > >
> > > 2) **Synthetic Trajectory Data:** When a model synthetically generates image trajectories to complete tasks, as in (https://developer.nvidia.com/blog/enhance-robot-learning-with-synthetic-trajectory-data-generated-by-world-foundation-models/), one needs to label these poses in order to physically achieve it. Similarly, this application requires exploring natural variations in the actuation available, and CRL is a major step to achieve it.

---

### Official Review · Reviewer_PMxf · 2025-10-31

**Soundness:** 2
**Presentation:** 2
**Contribution:** 2
**Rating:** 6
**Confidence:** 3

**Summary:**

This paper presents a weakly supervised framework for recovering robot joint angles from images without requiring fully labeled data. The approach uses interventional causal representation learning (CRL), treating joint angles as controllable latent variables that can be identified through distributional changes induced by joint-level interventions. The method employs a three-stage pipeline: (1) dimensionality reduction via an autoencoder, (2) score difference estimation using binary classifiers, and (3) latent space disentanglement via a second autoencoder with sparsity constraints. The authors evaluate ROPES on a robot arm in simulation, demonstrating successful disentanglement of joint angles and competitive performance against a supervised baseline.

**Strengths:**

1. The paper builds on solid theoretical foundations from score-based CRL and bridges the theory–practice gap by applying it to a robotics problem. As also discussed by the authors, the pose estimation problem has various applications within the robotics domain.


2. The empirical results are strong, and the evaluation is fairly comprehensive (with some limitations noted below), covering multiple conditions, ablations, and comparisons against a SOTA method.

**Weaknesses:**

1. While the authors claim their method is completely label-free (L161), I have doubts about this. The method requires knowing which joint was intervened on for each dataset, which creates a form of weak supervision. Moreover, in the linear calibration step, a small labeled dataset of ground truth samples is required. Additionally, the interventional distributions need to be sufficiently distinct, probably requiring careful design of the experiments by the authors. Overall, the claim of being entirely label-free and unsupervised is somewhat misleading, and I believe the supervision and labeling simply occur at a different level rather than through traditional annotations.

2. It would have been useful to see results obtained on a real robot. The method also assumes static backgrounds, so its performance in dynamic environments remains unclear.

3. The pose estimation results are not compared with any standard pose estimation algorithms or even classical methods. While such methods typically require detailed labels to train, including them for comparison would provide better insight into the proposed method’s capabilities.

4. The three-stage training process likely introduces significant computational overhead, which is not clearly discussed in the paper. The authors should include a comparison of computational costs against standard baselines.

5. The paper would benefit from a more explicit discussion of failure cases and limitations. Some example questions that could be addressed are provided below.

**Questions:**

1. Can the method handle simultaneous interventions on multiple joints, potentially reducing the data collection?

2. How does the method perform when some joints are occluded or not visible from any camera angle?

3. How robust is the method to dynamic backgrounds? Nothing too extreme, for instance, if one set of trajectories is collected during the day and another at night, or if objects in the background are moved between robot movements?

4. What are the challenges in carefully designing the interventions to ensure the distributions are sufficiently distinct?

---

> ### Author Response · Authors · 2025-11-21
> **Reply to Reviewer PMxf**
>
> **Clarification on ROPES’ supervision**: Thank you for bringing out the nuance. We will adjust some of the terms to accurately describe what we mean by labeled data and supervision. Specifically, by unsupervised, we strictly meant the lack of actual pose labels. Our understanding of the prior work is that forward kinematics of state space to image rendering or pose labels were necessary for disentangling/recovering the latent pose variables. Since the main components of ROPES do not use any such label, we used unsupervised under proper caution, explicitly stating exactly what is meant, such as in L157 (”except for the intervention/dataset labels”) and L202 (”Except for the knowledge that some specific joint has been distributionally intervened on, we require no pose labels.”), and it was made clear in “Section 3.2: Data Generation: Intervention via Manipulation” as well. Nevertheless, to avoid any potential overstatement, we change “completely label-free” in L161 to “pose-label free (aside from a small calibration set)”.
>
> **Linear calibration step is inevitable for direct comparison**: We recall that we report results in two metrics: (1) mean correlation coefficient (MCC) for raw CRL output, and (2) mean squared error (MSE) for CRL output post-calibration. The first metric (MCC) does not use any label and has been the standard evaluation measure in CRL. For the second metric (MSE), we kindly note that a simple calibration step is *inevitable* to align the scale of the learned representations. Otherwise, a direct MSE evaluation is not possible, as the “unit” of the target variable will be unspecified, e.g., inches vs. centimeters, and taking the difference between quantities with mismatching units would be meaningless.
>
> **Experiments on real data**: The field remains in an early developmental stage in which even simulation-based evaluations are relatively sparse and often constructed around highly stylized, contrived scenarios. The absence of standardized datasets, benchmark problems, and shared testbeds limits our ability to conduct real-world evaluation and to establish comparable, reproducible results across studies.
>
> **New experiments on Robosuite**: To address some of the reviewer’s comments on real data, we identified Robosuite simulator which is based on the MuJoCo physics engine to be a challenging test bed to try out ideas. This simulator provides higher resolution images (512x512 versus the previous 128x128). We have added additional experiments establishing successful disentanglement results (Figure 16 in Appendix L).  For instance, we have been able to disentangle Joint 1 with an MCC of 0.90, showing the promise of our method in even more scaled-up settings. For generating the data, we use two camera views that are 90% rotated along a specific axis.
>
> **State-of-the-art comparison**: Before addressing comparisons with the state of the art, we would like to invite the reviewer to first check our general response regarding the intended scope of this paper and its key contributions from a CRL perspective. We believe that perspective is important for evaluating the contributions. Following up on the notes in the general response, we would like to emphasize that while robotics has been a key motivating application for CRL, despite significant advances in CRL, there is still no study that establishes the relevance of CRL to a well-defined specific robotics application. Formalizing the robot pose estimation as a CRL problem is a significant contribution of our paper. Specifically, we demonstrate how to frame the well-defined pose estimation problem as a CRL task and show how to create diverse data aligned with theoretical necessities.
>
> Given this context, we note that our primary objective is not establishing that we are advancing the state-of-the-art on pose estimation. Nevertheless, we show that we can achieve a performance on par with one of the most recent approaches, with the additional advantage that we do not require any labeled image data.  We note that the RoboPEPP baseline we adopted is one of the most recent papers on the topic (appeared in CVPR 2025) and provides comparisons with the only other approaches that predict robot pose with unknown joint angles (HPE (Ben et al., 2024) and RoboPOSE (Labbé et al., 2021)), and reports stronger performance. In the revised draft, we have included a discussion (beginning of Section 4.4) on the relevance to other methods through shared comparisons with RoboPEPP.
>
> Ban, S., Fan, J., Ma, X., Zhu, W., Qiao, Y., & Wang, Y. (2024, September). Real-time holistic robot pose estimation with unknown states. In European Conference on Computer Vision (pp. 1-17). Cham: Springer Nature Switzerland.
>
> Labbé, Y., Carpentier, J., Aubry, M., & Sivic, J. (2021). Single-view robot pose and joint angle estimation via render & compare. In Proceedings of the IEEE/CVF Conference on Computer Vision and Pattern Recognition (pp. 1654-1663).

---

> ### Author Response · Authors · 2025-11-21
> **Reply to Reviewer PMxf**
>
> **Three-step training**: We recall the roles and complexity of three stages of our training:
> 1. Autoencoder-1: This step is only used for dimensionality reduction, performed with an MSE reconstruction loss and a simple model architecture (details of which are given in Table 3). As it doesn’t involve a complicated training process, it adds only a low computational cost to the total overhead. Most image processing pipelines including in diffusion come with a tokenizing step. This is akin to that.
> 2. LDR training: This step is trained to do only binary classification, performed with a binary cross-entropy loss and a simple model architecture (details of which are given in Table 4). This step, too, doesn’t involve a complicated training process. The only complexity arises from training a different LDR network for each joint, but the training of each network itself doesn’t incur a heavy cost.
> 3. Therefore, the main bulk of training complexity lies in the optimization of autoencoder-2. This means that in terms of overhead, our 3-stage pipeline does not add heavy overhead.
>
> **Discussion on failures**: We thank the reviewer for their feedback. We have included a discussion about the limitations and failures of the proposed method in Section 5, and discuss some examples that align with the questions the reviewer raised below.
> - Multiple joint interventions are not supported by the theory.
> - In practice, we observe that consistent occlusion of a joint in training (for instance, due to specific camera angle) does impede the recovery of the occluded joint angles.
> - Joints whose MCC values were worse showed more degradation when we test the model on images with changed brightness levels.
> - We also observed that when interventions are *starkly different*, the LDR model learns to distinguish them too easily. However, we observed that this is actually harmful for the disentanglement step. Our hypothesis is that, once the classification task is too simple, the LDR model does not have to learn the density ratio perfectly, which causes issues for the subsequent sparsity-based loss function.
>
> **Q1 - Multiple-joint interventions**: Thanks for the insightful suggestion. Using multi-target interventions is an active research topic in CRL literature. However, even for the more restrictive case of linear transformations,  theoretical guarantees for recovering each latent variable require $O(n)$ environments (see Varici et al. 2024). It is an open problem whether exact recovery is possible with fewer (stochastic) multi-node interventions. Thus, for the scope of this paper, we remained faithful to the theoretically proven case and used single-node interventions for nonparametric transformations.
> Burak Varıcı, Emre Acartürk, Karthikeyan Shanmugam, and Ali Tajer. Linear causal representation learning from unknown multi-node interventions. NeurIPS 2024.
>
> **Q2 - Occluded joint angles**: In Section 4.1, we discuss how interventions on joints 1,3,5 do not create visible enough changes from a single camera view. Not surprisingly, in this case, the recovery of these three joints is not successful. This is the exact reason we also use a two-camera setup in Section 4.2, to capture variations from and recover *all* joints.
> We also want to emphasize the difference in training vs. inference for occlusions. Intuitively, the training process requires diverse *visible* data, e.g., images where variations in the joints are clearly visible, for learning a good model. On the other hand, as Table 2 shows, the inference on occluded test data still shows remarkable performance.
>
> **Q3 - Dynamic backgrounds**: We took the model trained on two camera views that gave the results in Table 1 in the independent model case and we increased the brightness level by a factor of 1.25x.. We are happy to report that barring joints 3 and 5, all other joints have the MCC values which are roughly greater than 0.86. Joints 3 and 5 even with the original setting has worse values. We don’t see them worsening considerably when we change the brightness in the test. A sample comparison for the brightness change and the resulting Table on the test set after brightness change is discussed in the Appendix K. Please note that we did not train on these brighter samples and we only tested the models as the reviewer asked.
>
> **Q4 - “Sufficiently different” is a mild condition for interventions**. We would like to note that Assumption 1 ( interventional discrepancy) is not a restrictive condition for intervention design. It simply requires that the density ratio of the two interventions cannot be constant over a region of positive volume in the latent space. It holds for almost any pair of widely-used continuous distributions. For instance, it holds for any two Gaussians with distinct means. Tables 6 and 7 show the exact parameterization of the interventions used in our simulations.

---

### Official Review · Reviewer_vQzk · 2025-11-01

**Soundness:** 3
**Presentation:** 3
**Contribution:** 2
**Rating:** 4
**Confidence:** 3

**Summary:**

This paper proposes an unsupervised framework named ROPES that applies causal representation learning to robot pose estimation. The goal is to recover the interpretable latent factors such as joint angles, using only interventional distributions rather than supervised labels. The proposed method consists of two autoencoders and a log-density ratio estimator, trained jointly to enforce identifiability under interventions. Experiments on the Panda-Gym simulator show that the proposed method can recover causal latent factors and achieve competitive reconstruction accuracy compared with a supervised baseline of RoboPEPP.

**Strengths:**

1. The paper considers the usage of causal representation learning into the robotics domain, considering a popular and reasonable question of whether CRL can work in practice. This is an important and timely problem in the area
2. The proposed method is clearly presented, including theoretical insights and an end-to-end design with two autoencoders and a log-density ratio estimator. The math formulation is easy to follow.

**Weaknesses:**

1. The proposed method largely applies well-known score-based CRL results to the robot application. There is no significant theoretical or algorithmic advance beyond adapting the framework to robot pose estimation.
2. All the experiments are performed in the Panda-Gym system with grayscale synthetic images. However, the authors claim that they bridged the theory and practice. This is overstated without real visual data for validation. A small real-world test would be better to support this claim.
3. The experiments only include one baseline RoboPEPP (a supervised method), which is not sufficient. Meanwhile, this paper lacks ablations studies, e.g., removing LDR or sparsity constraints, and deeper analysis of learned latent representations. It remains unclear which components are crucial for performance and how interpretable the learned latents truly are.

**Questions:**

1. Is there any experiments on real-world tasks that apply ROPES to real image data?
2. How sensitive is ROPES to the choice of hyperparameters or encoder architecture?
3. Would it be feasible to extend ROPES to temporal settings with sequential observations?

---

> ### Author Response · Authors · 2025-11-21
> **Reply to Reviewer vQzk**
>
> **CRL versus robotics perspective:** We would like to thank the reviewer for thoughtful evaluation and questions. Before addressing specific questions and comments, we would like to invite you to first check our general response regarding the intended scope of this paper and its key contributions from a CRL perspective. We believe that perspective is important for evaluating the contributions.
>
> **Advances made by the paper:**
>
> **Contribution to CRL**: Following up on the notes in the general response, we would like to emphasize that while robotics has been a key motivating application for CRL, despite significant advances in CRL, there is still no study that establishes the relevance of CRL to a well-defined specific robotics application. Formalizing the robot pose estimation as a CRL problem is a significant contribution of our paper. Specifically, we demonstrate how to frame the well-defined pose estimation problem as a CRL task and show how to create diverse data aligned with theoretical necessities.
> Score-based method: The choice of score-based CRL adoption is deliberate. This is a framework with theoretical guarantees in complex nonparametric settings, such as robotics tasks. We also note that translating the theoretical elements of the score-based framework to a pose estimation involves several important steps. We emphasize that the objective of this paper is to advance understanding of the applications of CRL in robotics to complement the extensive recent theoretical advances. We believe that this is an important gap to address.
>
> **Comparisons**: Since it is critical, at the expense of being repetitive, we would like to note that this paper’s contribution is advancing the applications of CRL by formalizing that a well-defined robotics problem can be posed and solved as a CRL problem. Our primary objective is not establishing that we are advancing the state-of-the-art on pose estimation. Although, we also establish that we can achieve a performance on par with one of the most recent approaches, with the additional advantage that we do not require any labeled image data.
>
> **Theory-practice gap**: We entirely agree that there is still much to do to fully bridge the theory-practice gap in CRL. As stated in the abstract (L16-17), our paper represents *a step towards* this significant goal. To prevent the scope being misconstrued, we revise the other two mentions in the paper (on Page 3 and Page 9), from “bridges the gap..” to “takes a step towards bridging the gap..”
>
> **Additional experiments**:
>
> **New experiments on Robosuite**: To address some of the reviewer’s comments on real data, we identified Robosuite simulator which is based on the MuJoCo physics engine to be a challenging test bed to try out ideas. This simulator provides higher resolution images (512x512 versus the previous 128x128). We have added additional experiments establishing successful disentanglement results (Figure 16 in Appendix L).  For instance, we have been able to disentangle Joint 1 with an MCC of 0.91, showing the promise of our method in even more scaled-up settings. For generating the data, we use two camera views that are 90% rotated along a specific axis.

---

> ### Author Response · Authors · 2025-11-21
> **Reply to Reviewer vQzk**
>
> **Analysis and comparisons**:
>
> **Does CRL work for pose estimation?** Irrespective of whether the CRL approach outperforms the existing approaches, the central message we want to get across is that pose estimation can be formalized as a CRL problem. Our experiments show that the disentanglement of the robot pose variables, as guaranteed by theoretical results, is successful. This is demonstrated by strong results in the mean correlation coefficient (MCC) metric. The empirical observations are stronger than what the existing theory can guarantee. Specifically, the existing theory ensures identifiability of the latent variables only up to an invertible element-wise transform. In fact, this is known to be the best one can infer with interventional CRL, without making parametric assumptions (von Kügelgen et al. (2023, Proposition 3.1)). Our empirical results are stronger in the sense that we obtain competitive mean squared error (MSE) results after only an **affine** calibration step, implying that learned representations are actually recovered up to an affine transformation.
>
> **More ablation studies**. Thanks for the suggestions for more ablation studies. We varied the learning rate for the key second stage (where we impose the sparsification penalty), varied the batch size, and varied the sparsification and reconstruction loss weights. We then evaluated performance using MCC metric. Figures 13 and 15 in Appendix K show that it is certainly critical to choose an appropriate learning rate and to set the relative loss weights correctly for the second autoencoder. In particular, Figure 15 shows that just using reconstruction loss alone cannot perform disentanglement (MCC of less than 0.15), whereas using a properly weighted score-based loss, we can achieve a strong disentanglement (MCC over 0.90).
>
> **Performance comparisons**: Our empirical results show that successful disentanglement followed by a standard calibration step achieves strong performance at recovering the exact latents. Even though it is not our objective to provide a comprehensive comparison with the pose estimation literature, we note that the RoboPEPP baseline we adopted is one of the most recent papers on the topic (appeared in CVPR 2025) and provides comparisons with the only other approaches that predict robot pose with unknown joint angles (HPE (Ben et al., 2024) and RoboPOSE (Labbé et al., 2021)), and reports stronger performance. In the revised draft, we have included a discussion (beginning of Section 4.4)  on the relevance to other methods through shared comparisons with RoboPEPP.
>
> **Extension to Sequential Observations:** Thank you for the interesting question. Generalizing score-based CRL to sequential observations would require considerable technical innovation. Specifically, the key component of the framework is a differentiable objective that promotes score sparsity between two distributions. It is not clear how to incorporate the natural variations observed in sequential data into the existing formulation. This is an important open question for the CRL theory to address.

---

### Author Response · Authors · 2025-11-21
**Global Response**

We would like to thank the reviewers for their thorough and thoughtful evaluations.
Before addressing specific questions and comments, we would like to invite the reviewers to re-evaluate the contribution from a different perspective. Specifically, the contribution is at the crossroads of two independent domains: “robotics pose estimation” and “causal representation learning (CLR).” This creates two lenses for evaluating the contribution:

*1. CRL as a solution for robot pose estimation*

*2. showcasing an application of CRL through a well-studied problem.*

Our intention in this paper is to address the second one, motivated by the following: CRL has been the subject of intense investigation over the past three years. There have been major advances in understanding the theoretical principles. However, evaluating the applicability of CRL to well-defined real problems is far scarcer, especially in robotics-related applications, which have been a major driver of the field. Specifically, robotics applications have long motivated CRL (Schölkopf et al., 2021). However, to our knowledge, there is no well-defined robotics problem to evaluate CRL algorithms and frameworks. Thus, a key contribution of our paper is the formalization of robot pose estimation as a CRL problem (as emphasized in the contributions paragraph, pp. 3, L147).  Other demonstration is that label free disentanglement is possible using CRL techniques at larger scales in a problem of interest in the real world.

**Performance on pose estimation**: We deliberately chose pose estimation as the application domain, given the rich literature indicating broad interest in the problem. This has been a choice to avoid generating a contrived problem. From this perspective, we want to emphasize that our intention is not to claim that we are advancing the art of pose estimation. Having said that, we have provided comparisons with one of the most recent studies on pose estimation (CVPR 2025) and have shown achieving a performance on par with that of the state-of-the-art solutions with some advantages in the requirement (e.g., we do not need pose-image labeled data).

**Experiments on real data**: The field remains in an early developmental stage in which even simulation-based evaluations are relatively sparse and often constructed around highly stylized, contrived scenarios. The absence of standardized datasets, benchmark problems, and shared testbeds limits our ability to conduct real-world evaluation and to establish comparable, reproducible results across studies.

**New experiments on Robosuite**: To address some of the reviewer’s comments on real data, we identified Robosuite simulator which is based on the MuJoCo physics engine to be a challenging test bed to try out ideas. This simulator provides higher resolution images (512x512 versus the previous 128x128). We have added additional experiments establishing successful disentanglement results (Fig. 16 in Appendix L).  For instance, we have been able to disentangle Joint 1 with an MCC of 0.91, showing the promise of our method in even more scaled-up settings.

**More ablation studies**. We have performed these additional studies by varying the learning rate for the key second stage (where we impose the sparsification penalty), varying the batch size, and changing the sparsification and reconstruction loss weights. We then evaluated performance using the MCC metric. Figures 13 and 15 in Appendix K show that it is certainly critical to choose an appropriate learning rate and to set the relative loss weights correctly for the second autoencoder. Importantly, these ablations also highlight the role of the sparsity loss, whose importance is consistent with the theory.

---

### Meta-Review · Area_Chair_Z3ht · 2026-01-07

**Summary:**

The primary disagreement between the reviewers and the authors is the perspective. The reviewers judget the paper from a roboticist's perspective to evaluate the usefulness and performance of the method, which is lacking. The authors argue that the paper is an application of Causal Representation Learning and argue that the pose estimation problem can be formalized as a CRL task. I can understand both perspectives, but the reviewers concerns resonate more with me. If the authors want to focus on CRL as a framework for robotic problems and showcase the generality of the framework, they should try to show at least two major applications of CRL in addition to pose estimation. Showing suboptimal performance and without real-world experiments is a rather unconvincing way to present what the authors intended. Therefore, I recommend reject.

**Reviewer Concerns:**

Addressed Concerns:
The authors successfully mitigated concerns regarding overstatements of their methodology by clarifying that "unsupervised" strictly refers to the absence of pose labels, acknowledging the need for intervention labels and a small calibration set, and subsequently modifying the text to "pose-label free". They rectified issues with misleading performance presentation in Table 2 by removing bolded emphasis where the baseline was superior and adding data to demonstrate competitive performance in low-data regimes . Furthermore, the authors clarified that the computational overhead of their three-stage pipeline is minimal due to the simplicity of the first two stages and corrected the description of their occlusion experiments from "real-world corruptions" to "artificial occlusions" to avoid overclaiming robustness.

Outstanding Concerns:
The most significant outstanding friction involves the demand for real-world physical robot experiments; while multiple reviewers requested real visual data to bridge the "theory-practice gap".  A fundamental disagreement regarding novelty also persists, as reviewers characterized the work as a straightforward application of existing score-based CRL to a new domain. Additionally, requests for broader comparisons were only partially met; the authors declined to include classical pose estimation baselines, insisting that the single baseline used (RoboPEPP) is the most relevant comparison for unknown joint angle prediction, and they noted that multi-joint interventions remain theoretically unsupported.

**Reviewer Scores:**

I believe the reviewers will maintain their original reviews which lean towards negative.

---

### Decision · Program_Chairs · 2026-01-26

Reject